# Plant Nutrition—New Methods Based on the Lessons of History: A Review

**DOI:** 10.3390/plants12244150

**Published:** 2023-12-13

**Authors:** Martin Kulhánek, Dinkayehu Alamnie Asrade, Pavel Suran, Ondřej Sedlář, Jindřich Černý, Jiří Balík

**Affiliations:** Department of Agro-Environmental Chemistry and Plant Nutrition, Faculty of Agrobiology, Food and Natural Resources, Czech University of Life Sciences, 165 00 Prague, Czech Republic; asrade@af.czu.cz (D.A.A.); suranp@af.czu.cz (P.S.); sedlar@af.czu.cz (O.S.); cernyj@af.czu.cz (J.Č.); balik@af.czu.cz (J.B.)

**Keywords:** long-term field experiments, soil and plant analysis, modern nutrition methods, stressful environment

## Abstract

As with new technologies, plant nutrition has taken a big step forward in the last two decades. The main objective of this review is to briefly summarise the main pathways in modern plant nutrition and attract potential researchers and publishers to this area. First, this review highlights the importance of long-term field experiments, which provide us with valuable information about the effects of different applied strategies. The second part is dedicated to the new analytical technologies (tomography, spectrometry, and chromatography), intensively studied environments (rhizosphere, soil microbial communities, and enzymatic activity), nutrient relationship indexes, and the general importance of proper data evaluation. The third section is dedicated to the strategies of plant nutrition, i.e., (i) plant breeding, (ii) precision farming, (iii) fertiliser placement, (iv) biostimulants, (v) waste materials as a source of nutrients, and (vi) nanotechnologies. Finally, the increasing environmental risks related to plant nutrition, including biotic and abiotic stress, mainly the threat of soil salinity, are mentioned. In the 21st century, fertiliser application trends should be shifted to local application, precise farming, and nanotechnology; amended with ecofriendly organic fertilisers to ensure sustainable agricultural practices; and supported by new, highly effective crop varieties. To optimise agriculture, only the combination of the mentioned modern strategies supported by a proper analysis based on long-term observations seems to be a suitable pathway.

## 1. Introduction

Most yield-increasing steps related to plant nutrition were performed during the 20th century. New strategies, therefore, do not confer a strong yield increase but rather stabilisation and sustainability.

Several books and studies related to the plant nutrition were released recently. However, they are usually dealing with a detailed description of one or few scientific goals, e.g., molecular genetics [1], plant mineral nutrients related to molecular biology [1,2], physiology processes related to plant nutrition [3], fertiliser application strategies [4], and beneficial plant–microbe interactions [5]. However, more complete insight is almost missing due to the brightness of plant nutrition research area. Therefore, we prepared the brief overview of current challenges in plant nutrition at the start of the 21st century. The manuscript is divided into four main sections. The first highlights the importance of long-term field experiments, and the second focuses on new technologies and methods used to determine plant–soil–environment relationships, including data analysis. The third part continues with new fertilising strategies and/or the improvement of nutrient availability. Finally, the main environmental risks, such as biotic and abiotic stress, are described as the most yield- and quality-limiting factors.

Generally, there are many new possibilities to improve yield and product quality and long-term sustainability related to plant nutrition. However, with increasing food and feed demands, there is an increased risk of under- or over-fertilising, soil fatigue, different stressors, and soil contamination. It is impossible to summarise all new approaches to plant nutrition, as well as their risks. Therefore, in this review, we describe the methods and strategies we consider the most important for the near future. Furthermore, this review points out some new possibilities for potential researchers investigating plant nutrition.

## 2. Challenges of Long-Term Observations Related to Plant Nutrition

When studying modern pathways in plant nutrition, history cannot be forgotten. Some earlier validated data are often only repeatedly confirmed or refined in new studies. To obtain significant results, long-term observations are needed. Good results in field conditions of one or two years do not necessarily mean that the applied strategy will be successful in the following seasons. Therefore, we start this review with a section dedicated to long-term field surveys, which provide valuable information about the effect of different strategies in plant nutrition. 

Soil is used as a growth medium for a plant by providing water, oxygen, and nutrients [6]. However, the natural system is unable to supply sufficient nutrients for long-term plant nutrition to maintain successive and constant yields [7]. Soil texture, structure, biological conditions, bulk density, pH, electrical conductivity (EC), cation exchange capacity (CEC), and porosity are limiting factors for optimal plant and root growth and biological activity of the soil to release nutrients [6,8]. 

In the 21st century, the use of mineral fertilisers alone will not be enough to increase crop production worldwide; rather, another fertilisation scenario/scheme must be investigated to fill the mineral fertilisation gap, such as amendments with other organic fertilisers and management practices and breeding strategies for nutrient utilisation efficiency and other technological interventions [9]. In addition, the excessive application of mineral fertilisers also poses serious environmental problems, such as pollution, soil degradation, eutrophication of water bodies, and deterioration of ecological functions [10]. 

The three most important parameters having a direct influence on soil nutrient status and availability, together with long-term sustainability, are as follows: (i) fertilisation, (ii) crop rotation, and (iii) soil tillage systems. Below are some of the many proven examples: Fertilisation with organic and mineral fertiliser can alter the total content of soil organic matter, composition, diversity, microbe activity, and mineralisation of soil organic matter (SOM) in the order of no fertiliser (C0) ≤ mineral fertiliser (NPK) < mixture of mineral fertiliser and straw < farmyard manure (FYM) [11]. Similarly, long-term applications of both organic and mineral fertilisers also increase crop yields, nutrient uptake, and soil productivity in the order of organic fertiliser > 50% organic fertiliser and 50% NPK > NPK > mineral NP > mineral PK > mineral NK > C0 after 18 years of continuous fertiliser in the North China Plain [12]. Veum et al. [13] mentioned that the quantity and quality of soil organic matter increased in the order of FYM > NPK > C0 from a Sanborn (Columbia, MO, USA) long-term field experiment. The long-term application of NPK, FYM, or cattle slurry to an experimental field at Praha-Ruzyně, Czech Republic, for 60 years showed a significant difference in humic substance content and quality and a higher available nutrient content, soil reaction, and SOM stock [14].Long-term crop rotation has also increased agroecosystem diversity, improving crop yields and soil health; increased climate change adaptation or agricultural resilience; increased the water holding capacity and microbial diversity of soil, long-term yield gain, and reduction of biotic factors (plant pathogens, insect pests and weed pressure) [15]; and increased crop residue returns to soil by up to 100%, resulting in a large carbon and nitrogen stock in the soil [16].Different tillage practices also have a significant effect on SOM dynamics and carbon content within soil aggregates [17]. Zero tillage (ZT), reduced tillage, and organic farming systems provide a better preservation of soil aggregates and soil quality, high carbon and soil organic matter content, increased carbon sequestration capacity, maintenance of crop productivity, and more sustainable agricultural systems compared to conventional (CON) systems [17,18,19]. However, the amounts of SOM stocks, dissolved organic carbon (DOC), and microbial biomass carbon are significantly influenced by depth and tillage, being higher at a 0–20 cm depth under RT compared to CON [18]. Despite its potential benefits, ZT has a serious impact on developing herbicide-resistant weeds, weed seed production, and weed/crop competition [16].

Generally, according to the results of long-term field experiments with organic and mineral fertilisation trends, crop rotations/intercropping, fallowing, the release of new cultivars, and land management practices (tillage vs. ZT), it is possible to describe not only the changes in the nutrient status but also in the amount of soil organic carbon pools and organic matter content in the soil, which also has many co-benefits, such as improved soil quality; enhanced resilience to climatic extremes or productivity; increased soil C sequestration or the net uptake of CO_2_ from the atmosphere [20,21]; decreased mobility of soil pollutants; good composition and diversity of the microbes [11]; and increased soil quality parameters, such as bulk density, SOM, total nitrogen, and available P and K [8].

SOM stock plays a dual role in the mitigation of climate change by removing or sequestrating CO_2_ from the soil up to a depth of 40 cm and maintaining soil health [22], which reduces the net CO_2_ emission into the atmosphere. The sequestrating capacity of SOM depends on the inputs and soil depth. For instance, the incorporation of crop residue and manure can sequester 195 and 435 kg of C ha^−1^ year^−1^, respectively, at a depth of 0–30 cm [23]. SOM is the third most important global carbon sink (>2500 Pg) among atmosphere and plant photosynthesis carbon sinks that helps in CO_2_ decarbonisation from the atmosphere for climate resilience agriculture [24]. The “4 per 1000” initiative also promotes different land use management practices to increase the SOM stock by the rate of 0.4% per year, which is equivalent to annual CO_2_–C emissions from fossil fuels [22].

Long-term field experiments allow us to investigate not only the benefits but also the risks of different agricultural strategies. The results have shown that the main challenges of soil degradation are poor agricultural management practices, e.g., overcultivation and removal of crop residues, which lead to erosion, loss of soil organic carbon, and nutrient imbalance [18]. Another challenge is the inappropriate use and mismanagement of land resources, especially in Sub-Saharan African countries, where soil erosion and land degradation are primarily caused by human activities, such as deforestation, overgrazing, and unsuitable agricultural practices [8]. Other challenges include different sources of waste (urban and industrial origin) and biological pathogens. Frequently, the use of organic waste products (e.g., green waste, sludge, or municipal solid waste) as organic fertiliser or soil amendments are responsible for the long-term accumulation of contaminants and trace elements (Cd, Cr, Cu, Hg, Ni, Pb, and Zn) in the soil [25]. For instance, the content of trace elements in the soil and sewage sludge application rates are positively correlated, i.e., 1 tonne of sewage sludge application per hectare per year in soil increases the content of Hg, Zn, Cu, Pb, and Cd in the soil by 6.20, 619, 92.9, 49.2, and 0.500 µg kg^−1^, respectively. This leads to the bioconcentration of heavy metals in grains [26].

Since 1843, many long-term field experiments have been conducted all over the world in different soil and texture types and climatic conditions (rainfall and temperature) with a variety of cultivars and treatments for different lengths of time (Table 1). Some long-term field trials, such as the Broadbalk experiment, Morrow plots, the eternal Rye/Halle trial, and the Sanborn experiment, have results that are more than 100 years old.

As shown in Table 1, earlier established experiments are mostly focused on the evaluation of different organic and mineral fertilising treatments, where mostly FYM, NPK, and the incorporation of crop residues are investigated in different crop rotations, as well as in monocultures. A similar experimental design can be observed around the world. Later established experiments incorporate treatments corresponding with actual scientific goals. Examples include sewage sludge or biogas station fermentation residues. However, long-term field experiments with newly developed products, e.g., biostimulants or nanofertilisers, are logically missing. According to already published studies, it can be generally stated (or better confirmed) that (i) the application of FYM or composts in agriculturally used soils leads to the highest sustainability in terms of carbon sequestration; (ii) the combination of organic manures and mineral fertilisers can support high and stable crop yields, as well as overall sustainability; (iii) only the regular incorporation of crop residues has a positive influence on carbon storage; and (iv) mineral fertilisation itself can support a high yield, but its influence on soil organic carbon sequestration and long-term sustainability is questionable.

## 3. New Instrumental Analytical Techniques 

As mentioned before, the results of long-term field experiments usually comprise only the evaluation of yield, total soil organic matter content and bioavailable nutrient status. However, the possibilities for analyses, as well as the evaluation of the results, have strongly improved in the last few decades. Therefore, it is possible to perform the evaluation from many different perspectives, as well as to realise new approaches. In this section, we take a brief look at some of the new methods and technical equipment used to determine the content of elements or molecules in plants and soil and their interactions. In addition to new technologies, the importance of investigating specific soil areas, such as the rhizosphere, and determining soil enzymatic activity and microbial composition is further highlighted. However, older methods should not be forgotten, as they are often used for the comparison and verification of newer methods. Another benefit of using older methods is the relatively long history of results which allows for a comparison on a temporal scale and prediction of future development. 

### 3.1. Spectrometric and Chromatographic Methods

The methods mentioned in this section are not generally new. For example, the origin of chromatography dates back to the end of the 19th century. The reason we are mentioning these methods is their strong development in the frame of sensitivity and spectrum of measured substances in the last few decades.

Inductively coupled plasma–optical emission spectroscopy (ICP-OES) is now a commonly used tool for determining different elements in plant soil systems. A high level of energy is provided to a sample by plasma, and sample molecules are broken down into individual ions, where the electrons are excited by the energy. To return to a low-energy position, ions release emission rays at the characteristic wavelengths for each individual element involved, and these emission rays are subsequently detected using a spectrometer [70]. ICP-OES detection limits are in the range of ppb (μg L^−1^) [71]. Similar to ICP-MS, ICP-OES is sometimes laborious in terms of sample preparation, as samples must be digested in acid, usually nitric acid in combination with hydrogen peroxide, as presented by Korkmaz et al. [72] and Barros et al. [73]. This method can be used to determine the content of B, Ca, Fe, K, Mg, Na, P, Al, Cd, Co, Ni, Cr, S, Cu, Zn, Si, Mn, Mo, Pb, and Nd in plant and soil material [72,73,74,75,76,77]. Both methods are used in the multi-elemental analysis, and ICP-MS is recommended for measurements of trace elements over ICP-OES, as it has lower detection limits [71].

Inductively coupled plasma–mass spectrometry (ICP-MS) is a reliable method, but it is laborious and requires digestive chemicals and extensive sample preparation [78], which comes from the fact that different matrix components can cause interferences (mainly polyatomic ions having the same mass as analyte) and must be separated [79]. Usually, sample preparation lies in digestion in nitric, hydrochloric, or hydrofluoric acid. The basic principle lies in the high-temperature ICP source converting the atoms to ions, which are later separated according to their mass/charge ratio (*m*/*z*) and detected using a mass spectrometer. The ions that are typically discharged by ICP are positive M^+^ or M^2+^ [80,81]. The detection limit, which is measured in ppt (ng L^−1^) [71], has a high dynamic range over several orders of magnitude and is capable of measuring almost every element in the periodic table (except anions) and phase [82], making it a powerful tool in elemental analysis. It is commonly used to determine the content of P, S, K, Mn, Fe, Ni, Cu, Zn, As, Se, Mo, Cd, Hg, Pb, Sb, Mg, Co, Mo, Ti, Cd, Si, Al, and Sr in plant and soil samples [82,83,84,85,86]. 

The abovementioned spectrometric methods allow us to determine even the isotopes of nutrients, as well as other elements. Thanks to this option, there is a chance to precisely track the movements of individual elements in the environment, starting with the fertiliser and ending with transport and placement in the plant, e.g., von Blackenburg [87].

Chromatography is also a well-known technique that is often used in relation to plant nutrition. This principle is described, e.g., by Pravallika [88]. The sample enters a stream of inert transport liquid (liquid chromatography—LC) or gas (gas chromatography—GC), which is called a mobile phase. Then, it enters a division tube known as a column, which has a particular filling, known as a stationary phase. Chemical compounds travel in gas steam inside a column at different rates, depending on their chemical and physical properties, as well as interactions with the stationary phase. Compounds then leave the column at different times from each other, allowing for their identification and quantification. One common application of this method is discussed in a review by Sankaran et al. [89], who showed that gas chromatography can be used to detect diseases in plants, as this method is capable of the qualitative and quantitative evaluation of certain volatile organic compounds produced by plants. These can be indicative of certain types of diseases, as well as act as cues for interactions with other organisms [90,91]. Nahar et al. [92] published a review in which the use of chromatography was discussed for the analysis of naturally occurring pharmaceuticals in plants. It can also be used to determine pesticide contamination [93,94] and certain organic pollutants, including hormones and pharmaceuticals, as negative aspects of plant nutrition [94,95,96]. 

### 3.2. Tomography, Magnetic Resonance, and X-ray Fluorescence

An example of a new, very powerful tool is computed tomography (CT), which can be used to understand the nature and spatial configuration of soil components. CT uses a computer and a rotating source of X-ray beams to create detailed cross-sectional images of the soil [97]. As a beam of X-ray radiation passes through the material, it attenuates due to its interaction with constituent atoms [98]. This method is capable of quantifying soil components of different densities because the attenuation directly varies with the density of each component [97,98]. This method is useful in visualising and quantifying various characteristics of root growth [99], such as the surface area; total root volume, length, and diameter [97,99,100,101,102]; root density; root water content [102]; and root tortuosity [97,100]; however, it is not possible to quantify the mentioned characteristics for the fine root fraction of the root system [97,99]. It can also be used to determine the soil particle size and distribution [103,104], particle shape [105], and soil structure [106,107] and to visualise and quantify different characteristics of porosity and water flow patterns in soil at different saturation levels, such as the pore architecture (number, size, and distribution of pores) [103,107,108], preferential flow patterns, [103,107], pore network density, surface area, length density, tortuosity, distribution [109,110], and steepness of pores [108] and their connectivity [110]. CT also has its drawback, coming from the fact that X-ray beam attenuation is influenced by soil type [109,111,112], proximity of roots to organic matter or air-filled pores [99,107,113], and root water status [102,114]. The water content of organic samples can also influence the resulting image contrast, making the plant root cells in CT images indistinguishable with increased water absorption [114], which is even more problematic in peat matrix samples as opposed to soil matrices [97]. Water-filled pores can also influence the image [115] and cause root misclassification [112]. Zappala et al. [116] concluded that it is possible to distinguish roots in undisturbed soil when the soil moisture content is kept at field capacity.

Neutron-computed tomography (NCT) involves the detection of the absorbance of neutrons produced by neutron sources as they pass through the matrix [117]. Its resolution is lower than CT, and it can be used in cases where the matrix and observed object do not contrast well [118]. NCT is very sensitive to hydrogen [119]. It is possible to visualise root–soil–water dynamics at the microscale by using this method. Moradi et al. [120] and Zarebanadkouki et al. [121] managed to visualise real-time water dynamics; however, these studies were performed on mostly sandy soils (up to 90%) with very little to no organic matter content. Mawodza et al. [119] successfully used NCT to map the root system architecture in aggregated, heterogenous soils with moderate amounts of soil organic matter and to map the water distribution. However, NCT requires access to a nuclear reactor or high-energy particle accelerator [115].

Positron emission tomography (PET) is a method that was first used in medicine to provide images of brain activity and to map active and passive zones based on the idea that the distribution of radionuclides in certain areas could be accurately and quantitatively visualised based on the emission of positrons. It derives its name and fundamentals from a group of the decay of radionuclides (^15^O, ^11^C, ^18^F, and ^13^N), whose key property is a short half-life (2, 20, 110, and 10 min, respectively), and involves the emission of positrons and their crucial role in the biology and chemistry of living organisms [122,123,124]. However, ^18^F stands out, as it can be substituted for hydrogen in some molecules and is used as an apoplastic proxy for water because the sizes of fluorine and hydrogen are comparable, and, thus, they both behave and bond similarly. Additionally, ^18^F is preferred over ^15^O in water transport studies, as it allows the experiment to run longer [125]. At the cellular and biochemical levels, plants and humans are similar, which makes the use of PET scanners in plant research possible [126]. In addition, Hubeau and Steppe [127], Knoblauch and Peters [128], Thompson [129], and Sevanto [130] described the importance of mapping the metabolic activity of plants and their components with regard to the distribution of oxygen and sugar, their movement and its rate, the location of the source and sink, lateral and axial movements and their coordination, and xylem and phloem interactions. This method can even be used to study lateral water exchange during water transport towards the leaves [131,132], as lateral water exchange is important in transporting sugar, as well as in buffering sudden changes in xylem water potential [133]. It is also used to quantify carbon uptake, translocation from the leaves to the roots [101], root tips [134] and fruit [128,129,130], and the velocity of this translocation [135,136,137]. It is also suitable for studying the plasticity and sectoriality of stems after damage has been done to the phloem [132]. Ariňo-Estrada et al. [138] proposed a novel method in which isotope ^22^Na was successfully used to demonstrate the transport of salts in plants.

Recently, tomography has seen an increase in the variety of methods used, mainly focusing on the measurement of electrical resistivity in plants and soil. Mary et al. [139] used electrical resistivity tomography (ERT) to map the soil water content and mise-à-la-masse (MALM) to image where the plant root system is in effective electrical contact with soil at places that are likely to be the location of root water uptake. A combination of both methods provides complementary information. Weigand and Kemma [140] and Corona-Lopez et al. [141] used spectral electrical impedance tomography (sEIT) to successfully characterise plant root systems. An alternative proposed by Gribbe et al. [99] provides information about root growth with better spatial resolution; however, it suffers from the inability to quantify the finest root hairs and requires complicated software.

Magnetic resonance imaging (MRI) is based on nuclear magnetic resonance (NMR), which exploits an intrinsic angular movement of atomic nuclei called spin, lending some of them, such as ^1^H (protons), a weak magnetic moment. ^1^H is the most used nucleus in MRI, thanks to its high detection sensitivity and abundant presence in living tissues. Imaging contrast in plant tissue is based on differences in proton density [142,143]. This method is useful for the quantification of the circulation and velocity of water in the xylem, phloem [144,145,146], and leaves [145], as well as lipid quantification and visualisation [147,148], plant histology [149,150], belowground mass morphology [143,146], root system architecture (root mass, length, diameter, root number, and root tip number), growth angle, spatial distribution [143,151,152,153,154], tortuosity [153], and root water status [154]. MRI is also useful in the quantification of soil pores and organic material [155]. Metzner et al. [143] compared CT and MRI techniques in a pot experiment and found that a higher water content affected MRI less than CT. Furthermore, MRI can sometimes detect more lateral roots and has strong root-to-soil contrast. However, sometimes CT can identify thin roots better, while noting that CT is more widely used since the analysis cost is generally lower.

Energy-dispersive X-ray fluorescence (ED-XRF) is based on the principle of exciting atoms in the sample material by sufficiently energetic X-rays, which then produce element-specific X-rays, whose intensities are proportional to the concentration of respective elements. All signals are collected simultaneously by the detector [156,157]. The potential advantage of this method lies in the fact that a measurement can be performed on ground powder pressed into pellets without digestive reactions. Bamrah et al. [156] used this method to measure the Mn, Fe, Cu, Zn, Se, and K content in leaves and found a very strong correlation with their reference method, using inductively coupled plasma–mass spectrometry (ICP-MS) for all nutrients except of K. Jagadeesha et al. [158] used ED-XRF to determine the content of Rb, Sr, Ca, K, Zn, Cu, and Mn in medical plants and in soil to estimate the uptake and retention of the mentioned elements. Iftode et al. [159] conducted an experiment in which soil samples and plant root zones were analysed for trace elements (Co, Cr, Cu, Cd, Pb, and As). Swain et al. [160] used this method to measure Cr, Mn, Fe, Co, Ni, Cu, Zn, Se, Rb, Sr, Pb, K, and Ca in the roots of medicinal plants. The detection limits of this method are in the order of μg g^−1^ [71]. The technologies mentioned in Section 3.1 and Section 3.2 can be used to determine nutrients in bulk soil and plant tissues. However, to better understand the plant–soil relationship, it is important to use them to investigate specific areas, such as the rhizosphere.

## 4. Methods Related to the Specific Environments and Nutrients Relationship

While analytical methods were mentioned in the previous sections, the following sections focus on the environments where these methods can be applied, such as in the rhizosphere and microbial communities, or to the study of nutrient relationships.

### 4.1. Investigation of the Plant Rhizosphere

Recently, research has increasingly concentrated on the soil–root interface—the rhizosphere. The reason is the importance of the rhizosphere in nutrient acquisition and in protecting against stress and diseases. The interactions of soil and root systems are seen as an important factor in improving the productivity and environmental effects of agriculture [161]. Mary et al. [139] mentioned that it is impossible to achieve a thorough understanding of root configurations in space and their temporal evolution by using invasive methods. The following two techniques focus on the study of the rhizosphere: (i) rhizotrons and (ii) rhizoboxes.

Rhizotrons are devices used for root phenotyping and the study of root growth and architecture in soil in a nondestructive way, using a transparent container, in which root growth can be observed [162]. They can be made from small (minirhizotrons) plastic bags [163] and discs [162], composite material plates [164], acrylic panels [165] for laboratory measurements, plastic tubes for field experiments [166], or large windowed underground rooms used to study the root development of trees [167]. Martins et al. [164], Smith et al. [165], Louvieaux et al. [166], and Hall et al. [168] used their designs for root system architecture assessment and calculated evapotranspiration and the effect of P fertilisers. For small-scale laboratory trials, minirhizotrons are commonly used with (i) germination pouches [163] and (ii) cone-tainers [169]. Cassidy et al. [162] recommended the use of their novel plastic disc design for mini-rhizobox studies, as it allows for better root development than cone-tainers and provides a higher survival rate than germination pouches. The root system can be observed directly by the eye [162] or camera, where characteristics of the root system are analysed by software from collected images [164,165,166]. Rhizotrons are a viable method, as they are cheaper than modern MRI, CT, or other previously mentioned nondestructive methods (see Section 3.1) that require very advanced equipment [165].

Another method to study the root environment is through rhizoboxes, which are more focused on soil, roots, and microbial interactions. Rhizoboxes are usually small devices designed for laboratory/greenhouse experiments and are made of plastic or composite materials; they allow for root growth to be observed through a transparent layer and for different parts of the rhizosphere to be sampled for root, soil, microbial, or chemical analysis. Common applications are the assessment of root exudates [170,171,172,173], composition of bacterial communities and their changes [170,172,174,175], enzyme assays [174,175], gene abundance [172,174], root growth characteristics [175,176,177,178,179,180], nutrient status [175,176,180,181], and metal toxicity [171,173,182]. Tao et al. [171] combined the rhizobox method with ^13^C labelling to estimate CO_2_ assimilation and transfer to the soil rhizosphere in the form of root exudates that have an effect on cadmium solubility. Schmidt et al. [177] used ^15^N labelling paired with a rhizobox experiment and successfully studied root plasticity in response to localised resource patches. Since there is no real uniformity in rhizobox designs, Mašková and Klimeš [183] compared two types of commonly used rhizobox types (cubic 3D and flat 2D rhizoboxes) and found that root and shoot growth is not hindered in either of the variations. All of the methods have in common facts that they are laborious (at least in sample preparation) and time-consuming.

### 4.2. Soil Microbial Composition

Soil microbial biomass, community structure, and physiological activities are active components of terrestrial ecosystems, such as grasslands, shrubs, secondary forests, natural forests, and plantation forests [184]. Its structure, distribution, and function are significantly impacted by land-use changes, aboveground vegetation, the rhizosphere [185], soil depth [184], soil salinity, soil organic matter content [186], and climate change scenarios (elevated temperature and elevated CO_2_ concentrations) induced by anthropogenic factors [187]. The microbial distribution and community structure across a range of terrestrial ecosystems depend on soil physicochemical properties (pH, depth, textures, organic matter content, and soil moisture), plant species, climatic regions, developmental stages, and human land management [188]. In the following section, different methods are mentioned that specifically target microbial biomass and community composition or structure.

The soil microbial community and structure can be evaluated by using the phospholipid fatty acid (PLFA) biomarker method (Table 2) to analyse the microbial physiological status and properties of the soil microbial community (bacterial abundance, fungal abundance, and bacterial composition). Here, the soil sample is extracted in a single-phase mixture of citrate/phosphate buffer, methanol, and chloroform [189]. Microbial biomass can also be determined using fatty acid indicator biomarker signatures according to Holík et al. [190], i.e., fungal biomass (PLFAfun) by concentrations of polyunsaturated fatty acids (18:2ω6,9) and bacterial biomass (PLFAbac) by terminally branched fatty acids (i14:0, i15:0, a15:0, i17:0, and a17:0), monounsaturated fatty acids (16:1ω7t, 16:1ω9, 16:1ω7, and 18:1ω7), methyl-branched fatty acids (10Me-16:0, 10Me-17:0, and 10Me-18:0), and cyclopropyl saturated fatty acids (cy17:0 and cy19:0), where the prefixes, “i”, “a”, “10Me”, “cy”, and “ω” refer to iso, anteiso, methyl group on the tenth carbon atom, cyclopropane fatty acids, and omega carbon, respectively. The total lipids (nmol lipid g^−1^ dry soil) present in the soil sample, as an index of total microbial biomass, and chemically similar lipid indicators represent ecological groups of microorganisms (fungal/bacterial, Gram-positive (GP)/Gram-negative (GN) bacteria, arbuscular mycorrhizal (AM) fungi, and saprotrophic and ectomycorrhizal (SEM) fungi) [185]. The microbial biomass can be estimated from the concentrations of PLFA biomarkers, using a conversion factor [191]. 

The absolute abundance of bacteria and fungi can be determined using a molecular technique of 16S/18S rRNA gene methods to analyse the soil bacterial and fungal community composition at a depth of 0–10 cm from the surface, and a real-time quantitative polymerase chain reaction is used to quantify the abundance of bacterial and fungal gene sequences in the soil [189]. Both the growth and 16S/18S rRNA gene copy numbers of bacteria and fungi are significantly affected by salinity and incubation time. However, the addition of plant material or organic matter can increase the growth rate of bacterial and fungal strains, respectively [185]. 

According to Zhao et al. [184], microbial biomass and physicochemical properties (total C, total N, C/N ratio, total P, pH, organic matter, and soil moisture content) significantly decrease at a depth of 10–20 cm across the five vegetation ecosystems, namely grassland, shrub land, secondary forest, planted forest, and natural forest, because the PLFA concentrations of bacterial and fungal communities are affected by different land use types and the soil depth.

### 4.3. Importance of Soil Enzymatic Activity and Its Determination

Microbial (together with plant) activity is closely related to soil enzymatic activity. Soil is a biochemical reaction site due to this activity, where the decomposition of organic residues, transformation of soil organic matter, and mineralisation of nutrients for plant growth activities take place using a group of intracellular and extracellular enzymes secreted from different sources [193]. Soil enzymes are synthesised and secreted by soil microorganisms (bacteria and fungi) [190], plant root exudates, animals, and the decay of animals and plants and through the interaction of plants and microbes in the rhizosphere [194,195]. The root rhizosphere is a major hotspot area for the production of enzymes, and the activity and distribution of soil enzymes is greater than in bulk soil due to the active and passive release of rhizodeposits (secretions, mucilage, exudates, and sloughed-off cells) from various root zones, which enhance microbial activity and enzyme production [196,197].

Various soil enzymes are involved in C, N, P, and S nutrient cycles or transformations (Table 3), through the decomposition of cellulose, xylane, chitin, phosphate, organic sulphur, or urea [198,199]. These enzymes belong mainly to three enzyme classes: (i) oxidoreductases involved in oxidation–reduction reactions (dehydrogenases, laccases), (ii) hydrolases involved in breakdown reactions (amylases, cellulases, glucosidases, phosphoesterases, sulphatases, amidases, and ureases), and (iii) lyases involved in breaking covalent bonds (decarboxylases, dehydratases, and ammonia lyases) [198].

The activities of protease, urease, acid/alkaline phosphatase, β-glucosidase, cellulose, saccharase, and xylanase enzymes are significantly higher in organic farming compared to conventional farming due to the presence of rich microbial diversity, high soil organic matter content, soil respiration, soil moisture content, and microbial biomass of C and N under organic farming conditions [195].

Soil enzymes (extracellular) play an important role in mediating soil carbon and nutrient cycling, such as the transformation, mobilisation, and acquisition of nutrients in soil by plants [202,203]. These enzymes are also responsible for the active decomposition of carbon and nitrogen compounds and transformation and mineralisation of soil organic matter [196,204] and are used as an indicator of soil–microbe–root interaction activities [189] due to their fast responses to environmental changes, disturbances, and contamination [194]. The enzymes carry out various biochemical reactions to maintain biogeochemical processes in soil, e.g., the mineralisation of N, C, S, and P from organic compounds (chitin, organic sulphur, xylane, urea, cellulose, hemicellulose, and lignin), stabilisation of the soil structure due to the higher stability of soil aggregates [195], and degradation of very toxic compounds or heavy metal contaminates into nontoxic compounds by chelating them into complexes [198]. Soil enzymes play a significant role in the formation and transformation of metal oxides, biogeochemical nutrient cycling, maintaining soil fertility and health, and protecting the environment from pollutant molecules [195,199]. In addition, metal oxides and clay minerals have a great effect on soil biological properties [195].

Nevertheless, enzymatic activities and their composition in the soil are influenced by the physical (pH, moisture, and soil structure), chemical (CO_2_ and O_2_ status and bulk density), and biological (microbial biomass, fauna and flora, and soil organic matter) properties of the soil [192,199]; human disturbance (tillage, herbicide inputs, irrigation, farming practice, cropping, and fertilisation systems) [190]; heavy metal contamination; and high amounts of nutrients that reduce enzyme activities [198], degrees of land subsidence, soil depth, or temperature [193]. Soil pH affects soil microbial diversity and enzyme activities at different ranges (irreversible denaturation at a higher pH, more enzyme stability at an optimum pH, and reduced enzymatic activity at a low pH), but the stability of enzymes varies with the enzyme type and origin and depends on the soil pH shift [205]. The accumulation of excess elements, such as Ca^2+^, Mg^2+^, Cl^−^, SO_4_^2−^, Fe^2+^, and Mn^2+^, is responsible for the formation of calcified roots, Fe^3+^ precipitation, and plaque formation around the roots that leads soil enzymes to enter a dormant state [197]. These factors exert physiological stress on microbial and root activities and change the microbial biomass, community structure, adsorption capacity of enzymes by soils, rhizosphere pH, and chemistry [203]. Because of the importance of enzymes, many methods have been developed to measure the enzymatic activity in the soil or rhizosphere. Most are based on the use of substrates that are decomposed with specific enzymes. Enzymatic activity is subsequently determined based on substrate consumption [206,207,208]. 

### 4.4. Plant Nutrition Indexes—Calculation of Optimal Nutrient Status in Plants

As mentioned in previous sections, older and current methods often focus only on determining one variable for a specific time or place. However, plants, soil, and the environment are dynamic systems, where nutrients are often connected to each other even by synergistic or antagonistic processes. The nutrient uptake and content in plants can also dramatically change during vegetation periods. Therefore, proper plant nutrition is necessary to consider different nutrient relationships, as well as plant growth stages. This section is also dedicated to new investigation possibilities in this area.

Dilution curves for nutrients are calculated based on shoot biomass weight and shoot nitrogen concentration. The reason is that nitrogen, phosphorus, potassium, sulphur, and magnesium contents in the dry matter of whole plants substantially decline along with the progress of vegetation [209,210] and with the decreasing shoot nitrogen concentrations [211,212]. 

In the case of phosphorus, dilution curves calculated based on shoot nitrogen concentrations do not require the determination of shoot biomass, which makes it more practical for use in farmers’ fields [212]. Moreover, the critical phosphorus concentration in plant tissues (P_c_) of wheat calculated based on shoot biomass significantly differs among locations, whereas the location or region (Canada, Finland, and China) does not significantly affect this relationship if the P_c_ is described as a function of the shoot nitrogen concentration [213]. Djaman and Irmak [214] further extended this procedure by calculating the critical phosphorus concentration in maize shoot biomass in combination with the critical nitrogen model (P_c_ and N_c_), resulting in a more robust tool for determining the critical nitrogen and phosphorus in maize. Most of the calculations of the dilution curves cover the nutrient concentration determined in the whole shoot biomass. Belanger et al. [215] confirmed that predictive models of P_c_ as a function of nitrogen concentration in the uppermost collared leaves of wheat and maize could also be used for diagnostic purposes.

The calculation of the dilution curves enables the expression of a nutrition index (NI), which is the ratio of plant nutrient concentration to critical nutrient concentration (N_c_, P_c_, K_c_, S_c_, and Mg_c_) [209,216,217,218,219]. Nutrition indices are related to shoot biomass yield [220], leaf area index [221], grain yield [222], nutrition use efficiency [223], and quality of harvested products [218,224].

The aim of this section is not the nitrogen nutrition index due to the large number of studies focused on this topic, as stated by Lemaire et al. [219]. However, the same authors highlight the need for dilution curve computations in the case of phosphorus and potassium. Therefore, dilution curves of phosphorus, potassium, sulphur, and even magnesium are reviewed in terms of nutrition index expression. The precise monitoring of the uptake of these elements shows great importance due to the difficulties related to phosphate mining [225], the negative balance of potassium under conditions of insufficient organic fertiliser input [226], the effect of sulphur on nitrogen uptake by plants [227], and the quality of harvested products [228].

Ziadi et al. [229] pointed out that the use of the phosphorus nutrition index (PNI) alleviates the problem of variability caused by physiological age and environmental factors. Therefore, unlike the phosphorus shoot concentration, the use of the PNI diminishes the need to use critical phosphorus ranges. Liebisch et al. [230] stated that the correlation coefficients between the relative yield and the N:P ratio or PNI in grassland were higher than those between the relative yield and shoot phosphorus concentrations. The relative grain yield of wheat, rape, and maize increases with the increasing PNI values up to about 100 and then decreases, likely because of antagonism with zinc [230]. The relative potato tuber yields are also closely related to the PNI, validating the critical phosphorus dilution curve as a useful diagnostic tool for improving the phosphorus fertilisation of potato crop systems in the tropical region of Brazil [231]. Not only yield but also quality can be related to the PNI. Sedlář et al. [232] recorded a stronger correlation between the P/Zn ratio in the shoot biomass of wheat and PNI compared to the shoot phosphorus concentration, indicating the potential use of the PNI in terms of biofortification.

The results from long-term grassland experiments indicate that there is no simple positive relationship between the applied elements and their concentrations in plant biomass [233]. These findings are in accordance with the results of Sedlář et al. [232], who recorded a stronger relationship between bioavailable phosphorus in soil and PNI compared to phosphorus concentrations in the shoot biomass of winter wheat. Jouany et al. [234] and Garnier et al. [235] also recorded the consistency of PNI in grasslands with soil phosphorus availability. After 44 years of contrasted phosphorus fertilisation, Cadot et al. [212] confirmed a relationship between P_c_ and shoot nitrogen concentration for grain crops and the need to revise phosphorus fertiliser recommendations based on currently used soil phosphorus tests. In short, plant-based diagnostic methods can be used as an alternative or to complement soil analyses [236]. Mapping the PNI and potassium nutrition index (KNI) at the landscape or regional scale can provide valuable complementary information for soil survey systems [219].

The advantage of nutrition indices is their simple interpretation. Nutrition index values greater than 1 (or 100%) indicate excessive nutrient uptake, and values below 1 (or 100%) indicate nutrient deficiency [237]. According to Liebisch et al. [230] and Luna et al. [238], for phosphorus, an index of 0.8 (80%) corresponds to the critical value: a value < 0.8 (80%) indicates limiting nutrition for plant growth; an index value between 0.8 (80%) and 1.0 (100%) indicates that plants are in a situation of nonlimiting nutrient supply; and a value > 1.0 (100%) indicates luxury consumption. Fertilisation at the recommended rate results in a mean PNI between 0.8 and 1.2 (80% and 120%) [233]. Cadot et al. [212] recorded the highest relative grain yield of wheat and rape in the case of PNI = 0.8–1.2 (80–120%). The nitrogen nutrition index (NNI) or PNI values below 0.6 (60%) indicate severe nitrogen and phosphorus deficiency, which limits biomass production [233].

NNI, PNI, and KNI are more robust traits for quantifying crop nutrition status, and methods based on destructive sampling and lab analysis are mostly unsuitable to guide management decisions. Instead, these traits are best seen as benchmarks to calibrate other methods that are easier to implement and cost-effective [219,237]. Of course, phosphorus deficiency early in the growing season of plants cannot be easily alleviated with later phosphorus application. Therefore, producers can use PNI to adjust phosphorus fertilisation in the following growing seasons [213]. However, the relationship between the maize sulphur nutrition index (SNI) and chlorophyll metre reading at the V6 stage (around the sixth developed leaf) recorded by Carciochi et al. [239] provides potential corrections of plant nutritional status.

Liebisch et al. [230] confirmed the applicability of PNI as a phosphorus nutrition indicator in grasslands. However, in grasslands that include more than 20% legumes, a correction in the calculation of nutrition indices is required (NNI_c_, PNI_c_, and KNI_c_) to avoid overestimations due to the ability of legumes to fix atmospheric nitrogen [230,234,240]. When nutrient determination in grass fractions alone was not possible, Jouany et al. [234] recommended corrected nutrition indices according to the following equations: PNI_c_ = PNI + (0.5 × legume %)(1)
KNI_c_ = KNI + (0.5 × legume %)(2)
where PNI and KNI are nutrition indices determined for a mixture of species represented in the grassland, and legume % represents a visual estimate of the legume proportion in the mixture. 

Based on experiments with maize, Ciampitti et al. [211] and Carciochi et al. [239] stated that the dilution curve for sulphur is more attenuated than the nitrogen curve. Accordingly, for rape, the dilution rate for critical sulphur was lower than the nitrogen dilution rate, for which the N:S ratio in shoot biomass was not useful in predicting the nutritional status of crop growth and grain yield, but it did help predict the oil concentration, with a critical value of 7.7 [241]. These authors reported a critical SNI value of 0.74 for the maximum grain production of rape. The critical level of the sulphur nutrition index (SNI) necessary to achieve 95% of the relative grain yield of maize reached a value of 0.79 at the sixth leaf collar (V6) growth stage [239]. 

The calculation of the SNI based on the sulphur dilution curve expressed by Reussi et al. [242] proved to be a reliable indicator of the sulphur nutrition status of winter bread wheat. Both the optimal N:S weight ratio in shoot biomass and qualitative parameters of grain (particularly Zeleny sedimentation volume, grain protein content, and wet gluten content) were recorded when the SNI exceeded values of 0.8 at the beginning of stem elongation, 0.7 at the late boot stage, and 0.6 at the beginning of heading [218]. Even though the sulphur dilution curve for wheat was expressed by Reussi et al. [242] until the stage of visible flag leaf ligule, Sedlář et al. [218] recommended the calculation of the SNI at the beginning of the heading stage.

The calculations of critical curves are given in Table 4. Studies focused on increasing the rates of nutrients are preferred because, according to Belanger and Ziadi [243], increasing the nutrient supply should be a factor in obtaining reliable results.

## 5. Data Evaluation

The abovementioned extraction methods and analytical techniques allow us to obtain a lot of potentially interesting data. However, without adequate statistical analyses, the potential of these data can easily be lost. A proper statistical evaluation should follow each experiment. First, the experiment should be designed from the very beginning with a certain statistical method in mind. A method used to evaluate the results should be chosen during the designing phase of the experiment. Two of the most common methods used to evaluate results are the regression analysis and correlation analysis (providing 1,534,332 results when searching for “regression” and 2,254,586 results when searching for “correlation” in the Web of Science (searched 10 July 2023)). A regression analysis is used to determine the presence of a relationship between variables, while a correlation analysis is used to test its strength. The most abundant forms of the regression analysis are linear regression, nonlinear regression, least absolute deviation method, and nonparametric or polynomial regression. A correlation analysis estimates the strength of the relationships between variables. Some of the most widely spread methods are the Pearson, Kendall, and Spearman correlation analyses, each having its own characteristics. Other commonly used methods are the comparison-of-means models, which are based on comparing the mean value of one group of samples with a specific constant mean value (*t*-test) or second group of samples (two-sample *t*-test) or the same set of samples tested at different times (paired *t*-test). Some models, such as the analysis of variance (ANOVA; 125,734 results on Web of Science (searched 10 July 2023)) can compare multiple groups of data with each other. It is important to note that each of these methods has certain prerequisites that require the data to be tested for things such as the presence or absence of a normal distribution and homoscedasticity (tested, for example, via the Shapiro–Wilk test and Breusch–Pagan test, respectively).

## 6. The Ways of Modern Plant Nutrition

Low agricultural productivity, conventional fertiliser application systems, climate change, different biotic and abiotic factors, increasing consumption, and the costs of chemical fertiliser are bottlenecks for resilient agricultural systems around the globe. It is therefore obvious that new strategies must be investigated and developed in the near future (Figure 1). The following subsections build on the previously mentioned long-term experiments and the modern possibilities of their evaluation. Their main focus comprises prospective ways of increasing the yield and quality of production and its sustainability in relationship to plant nutrition.

### 6.1. Plant Breeding towards Better Nutrient Management

New information and technical equipment allow for the modification of the plants in terms of supporting nutrient acquisition due to breeding techniques. The main target strategies can be divided as follows: Changes in root distribution (supporting root hair growth) [250];Enhancement of symbiotic mycorrhiza [251];Supporting root exudate production [252];Physiological changes minimising metabolic nutrient requirements and changes in nutrient transporter behaviour [253];Root-mediated water transport to the soil surface, which increases the solubility of nutrients [254].

The negative result of some of these pathways is yield reduction. It is well known that the mechanisms for nutrient acquisition are better developed in wild plant varieties compared to cultivated varieties. The reason is that cultivated plants were bred to provide a high yield under a sufficient amount of nutrients. Wild plants can also better use the applied nutrients, but it costs them energy to produce a higher yield [254].

The reduction in phosphorus uptake by plants is an example of the potential in plant breeding. Phosphorus has become a limiting element in plant production due to its reduced natural sources [255]. Plants need phosphorus mainly to obtain energy for photosynthesis [256]. However, the phosphorus requirements for ATP and RNA production are relatively low. Bingham [257] estimated that, for 1 m^2^ of leaf area, only 0.12 mg P is needed. Optimally connected plant stands usually develop 6 m^2^ of leaf area per 1 m^2^ of soil surface [258]; thus, it is easy to calculate the necessary P uptake, i.e., approximately 7.5 kg ha^−1^ [259]. It is much less than the usual P uptake by plants, ranging between 15 and 40 kg P ha^−1^. This shows that a large amount of P is stored as a reserve in the form of orthophosphates, phospholipids, and phytates [260]. Some stored phosphates are needed for RNA synthesis [261], and some are required for phytin synthesis in seeds necessary for germination [262]. Plants also store more phosphorus than they use as a preventative measure in the case of a depleted P source or the interruption of its supply [259]. Some plant P also remains without use. Thus, the challenge in plant breeding is to reduce plant P uptake without further influencing yield and vitality [263,264,265]. This will not have a negative influence on the nutritional value of harvested products because monogastrics (including humans) cannot utilise P from phytates. Furthermore, phytates inhibit the utilisation of some essential nutrients (Ca, Mg, Fe, and Zn) [266]. Reducing excess P storage in vacuoles can also paradoxically improve the nutritional value of harvested plants. This is only one promising example of plant breeding towards the management of nutrient acquisition, but there are many studies with a focus on similar topics. Generally, new plant varieties are more dependent on precise human support to produce a sufficient yield of high quality. Therefore, new agricultural strategies are developed as mentioned in the following sections.

### 6.2. Precision Farming—Variable Nutrient Application to Save Costs and the Environment 

Variable-rate fertilisation (VRF) is the foundation and backbone of “precision farming”, which supplies reasonable agronomic chemicals, pesticides, herbicides, seeds, and sprays based on real-time demand, prescriptive maps, and crop phenotyping information in a site-specific manner [267,268]. Precision farming is a promising “agri-tech revolution” or “the era of smart farming approaches”, integrating holistic management systems, i.e., breeding × environment × management practices × high-throughput phenotyping tools, IT technology, variable-rate technology, and variable-rate application of agrochemicals for the efficient use of limited resources. Precision farming, also called “climate-smart farming”, can play a significant role in mitigating climate change [269] because the agricultural sector alone contributes about 13.5% of the total global anthropogenic greenhouse gas (GHG) emissions for climate change [270]. However, precision agriculture still has a low rate of adoption due to skill and expertise barriers, different perceptions among end-users, and high costs [271].

In precision farming, the field receives a variable rate of fertilisers according to the spatial farm management zone (soil types, landscape position, and management history) [268]. This variability arises from the field prescription map of soil, soil properties, water content, rainfall distribution, crop growth, chlorophyll content, canopy height, density, and biomass [271]. The ultimate objective of VRF is to improve nutrient uptake by crops, reduce GHG emissions, save costs and inputs, and reduce environmental impacts and nutrient losses [272,273]. There are important variable technologies applied to VRF to monitor the soil-nutrient status, yield, and harvest-related information [274]. High-throughput crop phenotyping technologies, such as remote sensing-based platforms (unmanned aerial vehicles (URVs), red-green-blue (RGB) cameras, normalised difference vegetation index (NDVI), and microwave back scattering) and map-based platforms (geographical information system (GIS), and global positioning system (GPS) maps) are used for variable-rate fertilisation prescription maps and site-specific crop and soil management [268,272,275,276,277]. For instance, NDVI and VRF are used to specify the amount of fertiliser and the location of the application [278]. Optical sensors (GreenSeeker’s and active canopy reflectance sensors) are used to estimate the N status in rice, soybean and corn [275]. Another wide range of spectra indices (i.e., NDVI, leaf area index (LAI), ratio vegetation index (RVI), chlorophyll index (SPAD), and NNI) is used to predict the N status and crop requirements based on real-time growth information [267,272]. Other remote sensing tools (latent space phenotyping) are also used to detect abiotic stress factors, such as drought, nitrogen deficiency, and salinity [277].

In general, precision farming relies on a variety of technology tools, such as VRT, VRF, IoT, GPS, remote sensing, algorism, modern machinery, and other high-throughput technologies. These technologies increase the accuracy of variable-rate fertiliser application and the efficient use of inputs; however, they are not yet widely adopted in small-scale farming systems [269]. 

### 6.3. Improvement in Nutrient Use Efficiency Due to Fertiliser Placement

Under- or over-fertilisation is always a problem in conventional farming systems. In conventional farming, uniform rates are used to treat all fields in a broadcast, either in top dressing or basal application, without prior information about the nutrient status of the soil or crop needs, whereas in precision farming, the variable rate is used to treat the field at a specific site within the field at the right time with an amount based on soil test information, map-based methods, and sensor-based map approaches [278]. This section is closely related to the previous one because precision farming and local application go hand in hand. The optimal dose for a variable-rate fertiliser application is regulated by different variable-rate technologies. Variable-rate fertiliser application and real-time variable-rate control technologies are being increasingly applied to precision farming to adjust the nutrient input–output requirements of crops based on “3S” technology (remote sensing (RS), GIS, and GPS maps) [268]. A wide range of examples of variable application techniques can be mentioned, e.g., map-based variable granular fertiliser applicator (placement of fertiliser in a crop row), sensor-based application based on real-time growth of NDVI and spectral vegetation indices (SVI) (identify N status based on chlorophyll content and canopy structure) [278], spin spreader (apply multiple fertilisers simultaneously), discrete element method (a dual-band fertiliser applicator), pop-up method (fertiliser seed coat) [279], or variable-rate fertiliser spreader (based on real-time growth (NDVI and SVI) information) [267]. Compared to a conventional fertilisation system, variable-rate fertiliser applicators save about 45% N fertiliser [278] and provide 26% of the economic benefits [280]. Variable-rate application is therefore cost-effective and high yielding, reduces environmental impacts, and increases uptake compared to uniform-rate application [281]. For instance, variable-rate application for maize plants saved 29–32% of fertiliser inputs and increased yield by 11–34% compared to the uniform rate [282,283]. Variable-rate treatment also saved about 40% and 38% of nitrogen fertilisers, using canopy-sized maps and yield prescription maps, respectively, and 23% of phosphate fertilisers [284]. In uniform-rate fertilisation, only 30% of the applied fertiliser is available for uptake, and the remaining 70% is subjected to leaching/sorption/runoff [281]. Baligar et al. [285] reported that the overall use efficiency of applied fertilisers by plants is estimated to be <50% for N, <10% for P and <40% for K in conventional farming. The variable-rate treatment increased the fertiliser use efficiency by including decision-making codes, pneumatic cylinders (regulate fertiliser rate), or an electronic DC motor (control the discharge rate) [278]. In China, the site-specific N fertilisation approach reduced the total N rate by 33.6%, increased the yield by 0.4 t ha^−1^, and improved the agronomy efficiency of N by 20% and N recovery efficiency by 16% in rice–wheat cropping systems [286]. The nutrient use efficiency of crops can be calculated using data on yield response, recovery, agronomic, internal nutrients, and physiological nutrient efficiency [283]. However, the crop yield and nutrient response vary significantly in response to climate change, variation in soil type, and landscape features in the field [268]. 

### 6.4. Biostimulants in Plant Growth and Nutrient Acquisition

One of crop production’s key factors is the achievement of effective plant nutrition and protection. However, commonly used agents can often contaminate the environment and threaten human health due to food chain pollution. It is also necessary to look for other approaches and strategies [287,288]. The use of pesticides and chemical fertilisers is a traditional method, but it causes negative side effects on the environment via the progressive resistance of pathogens to active substances or over-fertilisation. Recently, an effort has been made to develop harmless products based on microorganisms and active natural substances—biostimulants. It is expected that the development of new strategies will have a significant economic and environmental impact, particularly for future generations [259,289]. Biostimulants can contribute, depending on soil and climate conditions, to overcoming the barriers to nutrient availability. These compounds contain microorganisms (bacteria and fungi) and/or active natural substances (extracts from soils, composts or seaweeds, microbial residues, and plant extracts). Products are developed for a wide variety of plants (e.g., agricultural crops, grasses, and ornamental plants). Their effective use should cause the mobilisation of nutrients from less bioavailable forms in the soil [288] and support root growth [290,291] and mycorrhiza development [292]. Microorganisms may play an important role in enhancing P availability to plants and have been proven to enhance uptake directly by extending the root system (e.g., mycorrhizal associations), increasing orthophosphate mobilisation from soil organic and inorganic phosphorus, and stimulating root growth [293].

The most described bacterial biostimulants belong to different strains of *Bacillus* and *Pseudomonas*. However, the bright scale of other microorganisms has also been studied, e.g., *Paenibacillus*. Some positive effects of bacterial and fungal biostimulants are listed in Table 5. Bacterial biostimulants based on the *Pseudomonas* genus are a component of biofertilisers, and their use along with mineral fertilisers may serve as an effective approach for enhancing crop nutrient requirements, thereby leading to sustainable production. These beneficial microbes should promote plant growth by increasing nutrient availability when applied as a seed dressing or with foliar application. *Pseudomonas* can promote plant growth through both P solubilisation and biological nitrogen fixation. Furthermore, it can release Fe and Zn from hardly available forms [294,295]. 

Another microorganism—*Bacillus amyloliquefaciens*—produces a bright scale of metabolites, such as enzymes (e.g., chitinase, peroxidase or protease, and phosphatase), as well as different types of antibiotics (e.g., bacillomycin, fengycin, or difficidin). Therefore, it can support nutrient availability and protect plants against pathogens [296,297,298].

**Table 5 plants-12-04150-t005:** Biostimulants supporting nutrient acquisition and their influence on plant growth—updated based on the overview of Holečková et al. [299].

Bacteria	Experiment Conditions	Effect on Target Plant	Source
*Pseudomonas* sp.	Laboratory	Stimulation of tomato plants’ growth	[300]
	Pot; field	Nonsignificant effect on growth and nutrients uptake with maize	[301]
	Field	Higher grain and straw yield of barley	[302]
	Pot; field	Improvement of germination, growth, and yield parameters of maize	[303,304]
	Laboratory; greenhouse; field	Improvement of germination, root, and aboveground biomass length of maize	[305]
*Pseudomonas jesenii*	Greenhouse	Better growth of tomato plants	[306]
	Greenhouse; field	Higher aboveground biomass and grain yield of chickpea	
*Pseudomonas fluorescens and Cupriavidus necator*	Greenhouse	Co-inoculation led to promoted maize growth under drought stress	[307]
*Bacillus amyloliquefaciens*	Laboratory	Better root and shoot growth of maize	[298]
	Pot	Nonsignificant effect on growth and nutrients acquisition by maize	[308]
*Bacillus subtilis*	Field	Improvement of macro- and micronutrient uptake with tomatoes	[309]
	Field	Higher yield of aboveground biomass and roots of cabbage	[310]
*Bacillus cereus*	Field	Increased potassium-use efficiency and higher potato yield	[311]
*Lysinibacillus sphaericus*	Field	Increase in maize yield	[312]
*Paenibacillus mucilaginosus*	Pot	Improvement of trifoliate orange seedlings’ growth	[313]
	Pot	In combination with ash, it improved P mobility but immobilized NO_3_^−^ (experiments with maize)	[314]
**Fungi**			
*Trichoderma* sp.	Laboratory	Higher soybean yield	[315]
	Laboratory	Better growth of *Vigna unguiculata*	[316]
*Trichoderma harzianum*	Pot	Better germination and seedling growth of wheat	[317]
	Pot	Nonsignificant effect on nutrient uptake and growth of wheat	[301]
	Pot	Increased shoot and root length, dry mass, and grain yield of pigeon pea	[318]
	Pot	Higher acid phosphatase activity in soil (experiments with maize)	[319]
	Pot	Better growth of *Brassica juncea*	[320]
	Pot	Increased shoot and root length and dry weight, as well as yield of melons	[290]
	Greenhouse	Higher potato yield	[321]
*Penicilium Bilalii*	Rhizoboxes	Longer roots of maize	[293]
	Pot	Higher alfalfa yield	[322]
	Field	Higher grain yield of wheat	[323]
	Field	Longer roots and higher P content in pea roots	[324]
*Rhizophagus (Glomus) intraradices*	Greenhouse	Improvement of yield parameters of tea plants	[325]
	Field	Better growth of tomatoes	[326]

The most investigated fungal biostimulants have been strains of *Trichoderma*, *Penicillium* and *Rhizophagus* (previously known as *Glomus*). The genus *Trichoderma* comprises filamentous fungi that occur in most soil types and different habitats, and some strains have been successfully tested as biofertilisers. They act against a broad spectrum of plant pathogens. These fungi increase plant growth and development, as well as the development of the root system [290,291,317,327]. Furthermore, selected *Trichoderma* strains can improve plant nutrient uptake [328]. Increased growth also occurs due to its strong antipathogenic activity, hormone biosynthesis, improvement of nutrient uptake from the soil, and root development by increasing the metabolism rate of carbohydrates and increased photosynthesis [317].

Another promising fungal biostimulant is *Penicilium bilalli*, which is a soil fungus living in symbiosis with plant roots that release soil phosphates [298,329]. It has been successfully used as a seed inoculant for a variety of crops, e.g., wheat, maize, oilseed rape, bean, soybean, and legumes. This biostimulant supports the solubilisation of mineral phosphate and enhances plant P uptake [293,294] due to the production of acidic metabolites, such as citric and oxalic acids [330].

Biostimulants also belong to different extracts, mainly originating from seaweed and composts. Seaweed extracts are widely used as amendments in crop production systems due to the presence of nutrients and other plant growth-promoting compounds (amino acids, vitamins, cytokinins, auxins, and abscisic acid). They are reported to stimulate plant growth and yield [331]; enhance tolerance to environmental stress [332]; increase nutrient uptake; enhance antioxidant properties; and increase resistance against a broad range of pathogenic viral, bacterial, and fungal diseases, as well as insect attacks [333,334]. The most known and used alga is *Ascophyllum nodosum*.

Despite the mentioned positive effects of biostimulants, it is also necessary to provide a critical look at this topic. Most studies have originated from laboratory or pot experiments, where the plants were grown under controlled and often sterile conditions. Because of this, competition with other microorganisms is reduced. However, natural field conditions significantly decrease the effect of applied biostimulants mainly due to (i) competition with other microorganisms, (ii) changing weather conditions, and (iii) contrasting soil conditions compared to those where the microorganisms were isolated [308]. 

Therefore, the role of biostimulants is overestimated [335]. The main reasons are (i) the easier publication of significantly positive results; (ii) the bias of results by prevailing publications based on pot experiments; (iii) field experiments with extremely high (expensive) doses of biostimulants; and (iv) the advertising pressure of the producers, who are aiming to sell products.

As a concluding remark to this section, biostimulants have very good potential for plant protection when they are applied to foliage and do not come into direct contact with soil. However, the practical importance of biostimulants to mobilise nutrients is still questionable. The best results have been proven for greenhouse production, seed dressing, or local application. Despite intensive research in the last decade, we are at the beginning of our understanding.

### 6.5. Use of Waste Materials as a Source of Nutrients

The increasing world population goes hand in hand with the production of different waste materials, e.g., about 2.01 billion metric tonnes of municipal solid waste is produced annually worldwide. The World Bank [336] estimates that the overall waste production will increase to 3.40 billion metric tonnes by 2050. Only about 13.5% of today’s waste is recycled, and 5.5% is composted. Therefore, increasing pressure is being placed on waste recycling. One of the most suitable methods to use waste materials is plant nutrition. Some of them, e.g., compost extracts, can be considered biostimulants. However, there is a large group of typical waste materials with great potential for future plant nutrition, as well as carbon sequestration improvement.

Organic and industrial waste materials can be applied directly to soil or modified by different pathways. The main aim is to hold contained nutrients in the plant’s available form and use them to reach a higher yield and production quality [337].

The last two decades have strongly increased the agronomical use of organic materials previously deposited in landfills. Due to different processes (anaerobic fermentation, pyrolysis, burning, or composting), these materials are now applicable as fertilisers [338]. These materials usually contain high amounts of organic matter and nutrients, which can, together, improve soil fertility, yield, and product quality. There is still a lack of information about most of these fertilisers from a long-term point of view.

A very intensively studied material is biochar, which is a by-product of pyrolysis. For example, Wang et al. [134] showed the potential of farmyard manure biochar to act as a slow-release fertiliser. However, Huygens and Saveyn [337] pointed out inconsistent results in a meta-analysis of biochar experiments. It is also necessary to continue this research.

The ashes from burned plant biomass can also be used as a substituent of mineral fertilisers, but the quality of ash is strongly influenced by the burning process, subsequent procedures, and vegetation period of fertilised plants [337]. Wood fuel provides 40% of today’s global renewable energy supply, and thanks to its net carbon emissions being zero, its production is expected to grow in the future. Wood ash is also one of the cheapest and most available soil amendments in developing countries [339]. Soil application of wood ash supplies mineral nutrients, especially Ca, K, Mg, and P [340], and it increases the soil pH, with concomitant benefits such as element leaching reduction, mitigation of possible Mn and Al toxicities [341], and reduction of the heavy metal uptake by crops [342]. However, the solubility and, therefore, the plant availability of wood ash-bearing P are generally very low [341,343]. Therefore, there have been efforts to improve the bioavailability of nutrients by using different techniques, including the application of ashes with previously mentioned biostimulants, for instance, in the study of Mercl et al. [314].

An important source of nutrients is sewage sludge from wastewater treatments. The easiest and most commonly accepted method is its direct application to agriculturally used soil [344]. Directly applied sewage sludge is not only a source of bioavailable nutrients (mainly N and P) but also of organic matter, so its application also improves soil quality parameters [61,345,346,347]. The cost savings and possibility of quality improvement with composting are also advantages. However, the disadvantages include a high volume and long stabilisation in low-temperature conditions [348]. Another potential risk is the danger of environmental contamination due to trace elements, organic pollutants, parasites, and pathogenic microorganisms [349].

Nutrient recovery from sewage sludge is possible due to precipitation technology. It is one of the world’s most used techniques [350]. This process is the origin of struvites, hydroxylapatites, and calcium phosphates. This technology has already been applied in many wastewater treatment plants [351]. The lower solubility of precipitated materials in soils decreases the risk of environmental contamination with over-fertilisation [352]. Furthermore, the meta-analysis of Huygens and Saveyn [337] demonstrated that struvite has almost the same properties as mineral P fertilisers, with relatively consistent results independent of the grown plant and soil conditions.

Sewage sludge is also commonly treated with a burning process. The advantages of this technology are volume reduction and the destruction of pathogenic microorganisms and other organic pollutants as medicament residues. However, it also has many disadvantages: (i) the concentration of heavy metals increases, (ii) the availability of nutrients decreases, (iii) nitrogen and organic matter are lost, (iv) the costs for burning are usually high, and (v) a complicated application of sewage sludge ash into the soil. Because of these disadvantages, sewage sludge ash is more commonly used in the building industry as an additive to asphalt or cement mixtures [353,354,355].

The use of newer waste materials therefore presents a perspective for future agriculture. However, there are some limitations to their use, and these limitations were summarised by Withers et al. [259] as follows:Low quality of materials (even by a lower ratio of contaminants);Risk of environmental contamination (including eutrophication);Sociological aspects (e.g., direct application of sewage sludge);Negative influence on soil biochemical processes (mainly soil organic matter);Difficult long-distance transport.

### 6.6. The role of Nanotechnologies in Plant Nutrition and Water Holding in the Environment

Nanotechnologies as nanomaterials or nanostructures with different dimensions, sizes (ranging from 1 to 100 nm), origins (physical, chemical, and biological), compositions, and shapes are used as novel interventions to face various challenges in the areas of agriculture, plant nutrition and protection, environment remediation, and food product packaging [356]. 

Nanomaterials have unique physicochemical properties, e.g., a high surface-to-volume ratio, penetrability, reactivity, solubility, and afaster and higher translocation ability to different parts of plants make them more suitable for application [357,358]. Nanotechnology is also recognised as one of the six “Key Enabling Technologies” or “Nano-era” that contribute to sustainable growth and development in several fields of the European Commission [359]. 

Nanoparticles (NPs) can be fabricated from different materials, such as biological agents (fungi, bacteria, yeast, and plant extractants), semiconductors, metal oxides, polymers, ceramics, and agricultural residue, using a top-down approach by reducing large-sized materials to a nanoscale level and a bottom-up approach by adding small-scale levels of atoms and molecules to form nuclei and NP sizes [360].

The application of nanotechnologies (nanomaterials and NPs) in the agricultural industry brings the most promising potential opportunities and approaches for agri-tech revolution or Agriculture 4.0 (the era of smart farming approaches) [361] that enhance sustainable, efficient, and resilient agricultural systems by increasing precision farming techniques, such as enhancement with fertiliser, pesticide, and herbicide application, improving the use efficiency of necessary inputs by managing biotic and abiotic stress [249,362].

However, the overapplication of nanomaterials in soils may have ecotoxicity impacts on soil, aquatic and terrestrial ecosystems, phytotoxicity, and cytotoxicity (disruption of the microbial community) and can increase oxidative stress [356]. 

Nanofertilisers (NFs) are delivered encapsulated within nanomaterials, such as (i) NPs, (ii) nanotubes, and (iii) nanoporous materials; (iv) coated with a thin protective polymer film [363]; (v) as a particle; (vi) via emulsions of nanoscale dimensions; (vii) via nanocomposites containing macro- and micronutrients; (viii) via metallic nanoparticles; (ix) via microcapsules (slow release, quick release, pH release, moisture release, and heat release) [358,360,364]; (x) via carbon-based nanomaterials (graphene, carbon nanotubes, and carbon nitrides), which are used for the delivery of N, P, and K and essential nutrients (Fe, Zn, Cu, Mn, and Co) [365]; (xi) via natural biodegradable polymers, such as nanogels (chitosan and alginate used as a controlled release of N, P, and K), (xii) via nanoclay-based fertilisers (as zeolites, sorbents, or electrodes and a matrix of organic polymers of biological or chemical origin) to prevent undesirable nutrient losses and undesirable interactions of nutrients with soil, water, and air [358,362]; and (xiii) via metallic nanoparticles based on iron, zinc, and titanium oxides [366]. NFs are delivered to plants through key biological routes (root, soil-based; leaves, cuticle spray-based) at the right time, at the right dose, and in the right form, providing nutrients with high efficiency and low waste due to their faster and higher translocation ability to different plant parts [249,357,362,365].

NFs increase their nutrient use efficiency by three times with a targeted efficient release, increasing their stress tolerance ability (biotic—pathogens and abiotic; temperature; water stress; and flooding). They improve soil integrity, fertility, and aggregation by increasing organic matter, humic acid, and water content [249,358] and stimulating plant growth and development by enhancing elemental uptake. The use of nutrients and other beneficial elements (nano-SiO_2_, TiO_2_, carbon nanotubes, and nano-agrochemicals) increases absorption, water retention, and crop productivity [356,363]. Furthermore, NF application increases nutrient bioavailability during plant growth, which leads to a reduced loss of nutrients by leaching or runoff [364], reduction of waste, or lowering soil contamination by minimising nutrient inputs. NF application can increase the chlorophyll content (photosynthesis capacity) and overcome soil fertility-limiting factors regardless of the crop type [249,365].

## 7. A Stressful Environment as One of the Main Barriers to Plant Growth

This review contains a section describing brief stress conditions as key factors limiting agricultural sustainability. The abovementioned strategies can be limited by irreversible global changes caused mainly by human behaviour. Stressful environments (salinity, drought, and climate change) and the excessive use of agrochemicals and their interaction (antagonism effect) are the main threats to agricultural productivity and sustainability. Around 70% of crop survival, yield, and biomass production and 6% (800 million hectares) of the world’s land are negatively affected by major abiotic stressors (cold, heat, drought, and salinity). In particular, 20% of cultivated and 50% of irrigated land are directly affected by salinity [367].

Salinity is a major abiotic stressor based on natural conditions: (i) weathering parent rocks, thus discharging higher concentrations of soluble salts (such as sulphate, nitrate, carbonate, and chloride, as well as cations/anions); (ii) ocean salts carried with the rain and wind; or (iii) human origin (excessive chemicals, fertilisers supply, and irrigation). It is expected to affect 50% of all cultivated land worldwide by 2050 [368,369,370].

Soil salinity affects the following: Physical properties of the soil (soil pH, bulk density, and moisture),Chemical properties (osmotic stress due to a high concentration of Na^+^ and Cl^−^ and a reduction in K^+^ and Ca^2+^, ion imbalance or electrolyte leakage, ion toxicity, and low nutrient bioavailability),Biological properties (plant–pathogen interactions and microbial diversity) [369,371].

Salinity directly leads to soil compaction, drought stress, nutritional disorders, and the alteration of major plant physiological processes. Furthermore, stress can disturb metabolic pathways, leading to the accumulation or loss of metabolites, or it can be a reason for alterations in enzyme activities [368,372,373], the production of reactive oxygen species, the activation of antioxidant activities, and changes in the patterns of protein synthesis [367,374,375,376,377].

The excessive accumulation of salt ions (Na^+^ and Cl^−^) in the plant tissues also affects the dynamics and transformations of nutrients, i.e., uptake, translocation, and assimilation in crop plants, and it poses a water deficit or physiological drought stress in plant tissues, which ultimately leads to stunted growth, serious damage to the photosynthetic apparatus, a decrease in net photosynthesis, water gradient between root and soil, and reduced nutrient uptake [378,379]. 

Under salinity stress conditions, the uptake of Na^+^ and Cl^−^ ions increases in almost all cultivars and suppresses nutrient-related activities, such as metabolism, translocation, and partitioning, by imposing competition among nutrients and decreasing the content of N, K, P, S, Zn, Fe, Ca, and Cu in the soil solution, which leads to deficiency symptoms [196,369,370,380].

## 8. Conclusions

To reach high yields and production quality over a long-term trajectory, it is necessary to continuously modernise agricultural practices, including plant nutrition. However, without information based on historical data, modernising agricultural practices is not possible. Therefore, it is necessary to learn from the data, which provide us with long-term field experiments. Furthermore, a proper analysis has the same importance as data evaluation and interpretation. Only after considering these steps combined with modern analytical techniques (such as tomography, rhizosphere investigation, measurement of enzymatic activity, and microbial composition), as well as the further theoretical background of the farmers, proper fertilisation can be performed. Based on the abovementioned facts, this review briefly summarises some of the current strategies in modern plant nutrition:The breeding of new plant varieties with better nutrient acquisition or lower nutrient consumption could lower yields due to energy investments in root growth or metabolic processes. The second, lower-nutrient uptake strategy, can decrease plant vitality or the nutritional quality of harvested products.Precision agriculture allows us to save costs and, mainly, the environment due to the local application of fertilisers based on modern equipment and techniques. However, requests for precise input data interpretation and high input costs often discourage wider use.Biostimulants can improve nutrient acquisition. It is a promising technology for the future, but the results of field conditions are very inconsistent, and the published data often overestimate the influence of biostimulants.Fertilising with nonconventional waste materials (sewage sludge, struvite, ashes, biochar, and digestates) is necessary for sustainable agriculture. However, we must pay attention to nutrient bioavailability and the risk of environmental contamination with heavy metals and organic pollutants.The use of nanotechnologies is a promising method for holding water in soil and supporting nutrient bioavailability. The limiting factor is the high costs for use in agricultural practice.

It is also necessary to think about climate change and human influence on agricultural systems. With increasing demand for food production comes a higher risk of soil salinity, as well as drought and other stressors, caused by over-fertilising. Only by combining the mentioned plant nutrition strategies and avoiding different stressors can plant nutrition achieve long-term sustainability. 

## Figures and Tables

**Figure 1 plants-12-04150-f001:**
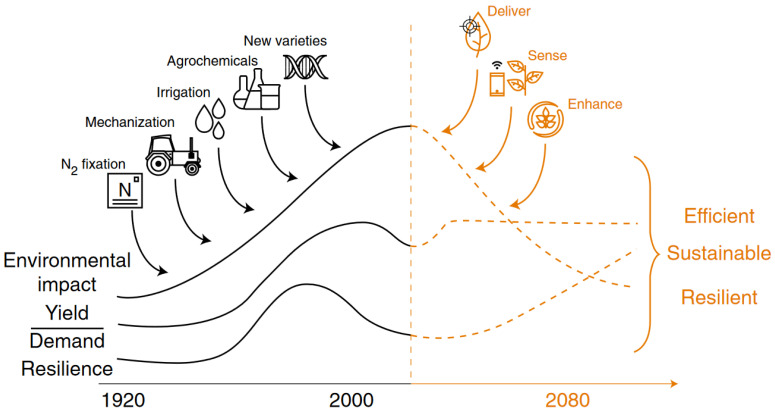
Agricultural revolution trends from a green revolution to a new agri-tech revolution to increase agricultural resilience and lower environmental impacts and make agriculture more sustainable to meet future food demands by the growing population adopted from Lowry et al. [249].

**Table 1 plants-12-04150-t001:** Overview of some selected long-term field experiments around the world.

Country	Location	Since	Founders	Main Focus	Soil Texture	References
United Kingdom	Broadbalk, Rothamsted	1843	Lawes and Gilbert	Effects of N, FYM, and straw on SOM/SOC stock	Clay	[22,27,28]
France	Essai Deherain	1875	Royal Agronomic Institution of Grignon	To compare the effects of organic amendments and inorganic fertilisers on crop yield	Agrudalf	[29,30]
USA	Morrow plots, Illinois	1876	GE Morrow	Investigation of soil clay mineralogy and its evolution upon agricultural practices	Silt loam	[31,32]
Germany	Halle/Saale Eternal Rye	1878	Julius Kühn	Effects of organic and mineral fertilisation	Loamy sand	[33,34,35,36]
USA	Sanborn Field, Missouri	1888	JW Sanborn	Effect of crop rotation and the application of FYM on crop production	Loam	[13,37,38]
Denmark	Askov—Lermarken	1894	Danish Institute of Plant and Soil Science	Effects of animal manure and mineral fertilisers on the content of C and N	Sandy loam	[39,40]
Poland	Skierniewice	1921	Warsaw University of Life Sciences	Effects of long-term nitrogen fertilisation on soil sorption capacity	Sandy loam	[41,42]
Norway	Moystad	1922	Bioforsk	Effects of FYM and mineral fertiliser on crop yield, nutrient supply, and changes in soil properties over long periods of time	Loam	[43,44,45]
Japan	Yamagata	1926	YIARC	Effects of long-term application of organic matter combined with inorganic fertilisers on stable carbon isotope	Inceptisol	[46]
Czech Republic	Praha- Ruzyně	1955	Crop Research Institute	Effects of organic and mineral fertilisation on soil chemical properties	Clay Loam	[14,47,48,49,50]
Sweden	Ultuna	1956	Swedish University of Agricultural science	Response of soil organic carbon to selected organic and inorganic fertiliser treatments	LoamY	[51,52]
Burkina Faso	Saria	1960	AfNet and TSBF	Application of inorganic (NPK) and organic fertilizers (FYM, crop residues, agroforestry, and tree biomass), and rotations and intercropping with grain legumes (cowpea, soybean, and groundnuts)	Tropical Soil	[51]
Italy	Padova	1962	University of Padova	Effects of recommended management practices on the SOC change	Clay, sand and peaty	[53,54,55]
Belgium	Gembloux	1974	Agricultural Research Center of Gembloux	Evaluation of long-term efficiency of livestock effluent on arable land as a source of nitrogen fertiliser	Sandy Loam	[56]
Kenya	Kabete	1976	NARL	Maintaining and improving the productivity of the soils through repeated use of inorganic fertilisers (in nitrogen and phosphorus), FYM, and crop residues under continuous cropping	Loamy	[57]
Niger	Sadore	1982	TSBF network	The effect of crop residue on the soil organic carbon and protection against erosion	Sandy	[57]
Hungary	Keszthely	1983	G. Láng	Effects of organic and mineral fertilisation	Loamy	[58,59]
Rep. of Serbia	IOSDV Novi Sad	1984	Institute of Field and Vegetable Crops	Evaluate the effects of crop residue on SOC stock	Clay loam	[23,60]
Spain	IOSDV Madrid	1985	CSIC	International long-term organic nitrogen fertilisation experiments	Loamy clay	[35]
Czech Republic	Červený Újezd	1992	CZU in Prague	Long-term application of organic (FYM, sewage sludge, and cattle slurry) and mineral fertilisers based on unified N dose	Loam	[61,62]
Austria	IOSDV Fuchsenbigl	1986	AGES	Investigate the effect of mineral N fertiliser in combination with selected organic amendments (FYM, crop residue, and slurry)	Loam	[63,64]
China	Fengqiu State	1989	Academy of Sciences	Investigate the effect of long-term fertilisation practices on soil productivity	Sandy loam	[12,65,66]
Slovenia	IOSDV Rakican	1992	E. Von Boguslawski	International long-term organic nitrogen fertilisation experiments	Loamy sand	[35]
Turkey	Cukurova	1996	Cukurova University	Effect of long-term mineral fertilisation, organic matter application, and mycorrhizal inoculation on some soil physical properties	Clay	[67,68]
China	Xianning	1998	Huazhong Agricultural University	Effect of long-term fertilisation on mineralization of soil organic carbon	Loam	[11,69]

Notes: YIARC, Yamagata Integrated Agricultural Research Center; AfNet and TSBF, African Network for Soil Biology and Fertility (AfNet) and of the Tropical Soil Biology and Fertility (TSBF); NARL, National Agricultural Research Laboratories; CSIC, Institute of Agrarian Sciences; CZU, Czech University of Life Sciences; AGES, Austrian Agency for Health and Food Safety; FYM, farmyard manure; SOM/SOC = soil organic matter/carbon.

**Table 2 plants-12-04150-t002:** Characteristic ester-linked fatty acids in the lipids of common soil biota adopted from Kandeler [192].

Fatty Acid Type	Frequently Found Biomarkers Signature	Lipid Fraction	Predominant Origin
**Saturated**			
Iso/anteiso methyl-branched	i, a in C14-C18	PLFA	GP bacteria
10-Methylbranched	10ME in C15-C18	PLFA	Sulphate reducing bacteria
Cyclopropyl ring	cy17:0, cy19:0	PLFA	GN bacteria
Hydroxy substituted	OH in C10-C18	PLFA	GN bacteria, actinomycete
**Monosaturated**			
Double bond C5	16:1ω5	PLFA/NLFA	AM fungi, bacteria
Double bond C7	16:1 ω7	PLFA	Bacteria widespread
	18:1 ω7	PLFA	Bacteria, AM fungi
Double bond C8	18:1 ω8	PLFA	Methane-oxidizing bacteria
Double bond C9	18:1 ω9	PLFA	Fungi
		PLFA	GP bacteria
	20:1 ω9	PLFA	AM fungi (Gigaspora)
**Polysaturated**			
ω6 family	18:2 ω6,9	PLFA	Fungi (saprophytic, EM)
	18:3 ω6,9,12	PLFA	Zygomycetes
	20:4 ω6,9,12,15	PLFA/NLFA	Animals widespread
ω3 family	18:3 ω3,9,12	PLFA	Higher fungi
	20:5 ω3,6,9,12,15	PLFA	Algae

Notes: NLFA, neutral lipid fatty acid; PLFA, phospholipid fatty acid; i, iso; a, anteiso, ME, methyl; OH, Hydroxy group; cy, cyclopropyl; GP, Gram-positive, GN, Gram-negative.

**Table 3 plants-12-04150-t003:** Examples of soil enzymes involved in nutrients cycling in soil.

Type of Enzymes	Sources of Enzyme	Substrate Acted on	Product	Type of Nutrient Released	Sources
Peroxidase	fungi	Petroleum, lignin, and ROS	hydrogen peroxide	N and C cycling and detoxification	[200]
β-1,4-glucosidase (BG)	MOS	β-glucosides	Glucose and cellobiose	C cycling	[196]
β-1,4-N acetylglucosaminidase	MOS	Chitin and peptidoglycan	Chitooligomers	N cycling	[196]
Cellulase	MOS (fungi and bacteria)	Cellulose and lichenin	Glucose	C cycling	[192]
β-xylosidase (BX)	Fungi	Xylane	Xylose	C cycling	[196]
Urease	MOS, animal, and plant	Urea	Carbonic acid and ammonia	N cycling	[193]
Protease	MOS and plant	Protein	Amino acids	N cycling	[192]
Alkaline/alkaline phosphatase	MOS (fungi and bacteria)	organic phosphorus	Orthophosphate	P cycling	[200]
Phosphodiesterases	MOS (fungi and bacteria)	Nucleic acids and another organic P	Orthophosphate	P cycling	[200]
Dehydrogenases	Soil bacteria	CO_2_, organic acids, and alcohols	Oxidized or reduced products	Proton transfer	[198]
Arylsulphatase or sulphatase	MOS, plant, and animal	Phenol sulphate and organic sulphate ester	Phenol and sulphate	S cycling	[198]
Deaminase	MOS (fungi and bacteria)	Amino acid	Ammonia and organic acids	N cycling	[198]
β-Glucosidase (BG)	MOS, animal, and plant	Cellobiose	Glucose	C cycling	[201]
Chitinase	Plants and MOS	Chitin	Carbohydrates and inorganic nitrogen	C and N cycling	[201]

Notes: ROS—reactive oxygen species; MOS—microorganisms.

**Table 4 plants-12-04150-t004:** Nutrition dilution curves for phosphorus (P_c_), potassium (K_c_), sulphur (S_c_), and magnesium (Mg) for wheat, maize, timothy (*Phleum pretense*), grassland, potatoes, soybean, and oilseed rape.

Crop	Region	Critical Nutrient Concentration	Reference
Wheat ^1^	CAN	P_c_ = 0.94 + 0.107 N	[229]
Wheat ^2^	CAN	P_c_ = 1.70 + 0.092 N	[229]
Wheat ^3^	CAN	P_c_ = 0.02 + 0.106 N	[215]
Wheat ^3^	CAN	P_c_ = 0.29 + 0.073 N	[215]
Wheat	CHE	P_c_ = 0.88 + 0.083 N	[212]
Wheat	CHE	P_c_ = 0.291N − 1.557 − 0.004 N^2^	[212]
Wheat	CHE	P_c_ = 4.44 × W^−0.41^	[212]
Wheat	ARG	S_c_ (%) = 0.37 × W^−0.169^	[242]
Wheat	CAN, FIN, CHN	P_c_ = −0.677 + 0.221N − 0.00292 N^2^	[213]
Maize	USA	P_c_ = 1.00 + 0.094 × (34.0 × W^−0.37^)	[214]
Maize ^4^	PLN	P_c_ = 3.2234 × W^−0.086^	[209]
Maize ^5^	PLN	P_c_ = 3.5191 × W^−0.085^	[209]
Maize ^1^	CAN	P_c_ = 1.25 + 0.104 N	[236]
Maize ^2^	CAN	P_c_ = 1.00 + 0.094 N	[236]
Maize	USA	P_c_ = 7.8 × W^−0.18^	[211]
Maize	CHE	P_c_ = 0.39 + 0.083 N	[212]
Maize	CHE	P_c_ = 3.49 × W^−0.19^	[212]
Maize ^3^	CAN	P_c_ = 0.82 + 0.089 N	[215]
Maize ^3^	CAN	P_c_ = 1.04 + 0.084 N	[215]
Maize ^3^	CAN	P_c_ = 0.003 + 0.082 N	[215]
Maize ^3^	CAN	P_c_ = 0.002 + 0.1011 N	[215]
Maize ^6^	USA	S_c_ = 7.0 × W^−0.30^	[211]
Maize ^7^	USA	S_c_ = 6.1 × W^−0.26^	[211]
Maize ^8^	USA	S_c_ = 5.7 × W^−0.24^	[211]
Maize ^3^	ARG	S_c_ = 2.13 × W^−0.23^	[239]
Maize ^4^	PLN	Mg_c_ = −0.221 × ln(W) + 2.2853	[209]
Maize ^5^	PLN	Mg_c_ = −0.225 × ln(W) + 2.502	[209]
Maize ^4^	PLN	Mg_c_ = 2.3014 × e^−0.004×W^	[209]
Maize ^5^	PLN	Mg_c_ = 2.4521 × e^−0.003×W^	[209]
Maize	USA	K_c_ = 88 × W^−0.21^	[211]
Maize ^4^	PLN	K_c_ = 37.41 × e^−0.006×W^	[209]
Maize ^5^	PLN	K_c_ = 39.231 × e^−0.005×W^	[209]
Timothy	CAN	P_c_ = 1.07 + 0.063 N	[243]
Timothy ^9^	CAN	P_c_ = 3.27 × W^−0.20^	[243]
Timothy ^10^	CAN	P_c_ = 5.23 × W^−0.40^	[243]
Grassland	FRA	P_c_ (%) = 0.133 + 0.091 N	[244]
Grassland	FRA	P_c_ (%) = 0.45 × W^−0.30^	[244]
Grassland	FRA	P_c_ (%) = 0.13 + 0.06 N	[244]
Grassland	FRA	P_c_ (%) = 0.15 + 0.065 N	[245,246]
Grassland	FRA	K_c_ (%) = 6.70 × W^−0.414^	[244]
Grassland	FRA	K_c_ (%) = 4.40 × W^−0.30^	[244]
Grassland	FRA	K_c_ (%) = 1.40 + 0.50 N	[244]
Grassland	FRA	K_c_ (%) = 1.6 + 0.525 N	[245]
Potato ^11,12^	COL	P_c_ (%) = 0.536 × W^−0.186^	[221]
Potato ^12,13^	COL	P_c_ (%) = 0.523 × W^−0.199^	[221]
Potato ^11,12^	COL	P_c_ (%) = 0.39 × LAI^−0.082^	[221]
Potato ^12,13^	COL	P_c_ (%) = 0.41 × LAI^−0.090^	[221]
Potato ^12^	ARG	P_c_ (%) = 3.919 × W^−0.304^	[217]
Potato ^11,12^	COL	K_c_ (%) = 8.84 × LAI^−0.437^	[231]
Potato ^12,13^	COL	K_c_ (%) = 5.95 × LAI^−0.149^	[231]
Potato ^12^	BRA	K_c_ (%) = 5.54 × W^−0.317^	[247]
Potato ^11,12^	COL	K_c_ (%) = 9.02 × W^−0.269^	[221]
Potato ^12,13^	COL	K_c_ (%) = 6.58 × W^−0.135^	[221]
Soybean	ARG	S_c_ = 2.8 × W^−0.11^	[210]
Rape	CHE	P_c_ = 5.18 × W^−0.39^	[212]
Rape	CHE	P_c_ = 1.67 + 0.657 N	[212]
Rape	CAN	P_c_ = 1.74 + 0.024 N	[248]
Rape	URY	S_c_ (%) = 1.22 × e^−0.18×W^	[241]

Notes: Nutrient concentrations are determined in whole shoot biomass if not defined otherwise. Critical nutrient concentration is expressed in g kg^−1^ (without units) and as a percentage (%), respectively. The dilution curves are expressed based on nitrogen concentration (N), shoot biomass (W), or leaf area index (LAI). ^1^ N nonlimiting conditions, ^2^ N limiting conditions, ^3^ nutrient concentration determined in uppermost collared leaf, ^4^ cultivar Pallazo, ^5^ cultivar Paroli stay green type, ^6^ without N fertilisation, ^7^ 112 kg N ha^−1^, ^8^ 224 kg N ha^−1^, ^9^ older swards, ^10^ younger swards, ^11^ cultivar Capiro, ^12^ nutrient concentration determined in shoot and tubers, ^13^ cultivar Suprema.

## Data Availability

Not applicable.

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
