# Peer review of "Plant Nutrition—New Methods Based on the Lessons of History: A Review"

_plants, 2023, doi:10.3390/plants12244150_

Round 1

Reviewer 1 Report

Comments and Suggestions for Authors

The manuscript can be published as it was written. The review is adequate with high contribution in the field of plant nutrition.

Author Response

Dear Reviewer 1

We are very grateful that you recommended direct publishing of our manuscript. We made the changes according to the two remaining reviews and we hope that the overall quality of the manuscript was improved. Attached is the tracked as well as clean version of corrected manuscript.

Yours sincerely

Martin Kulhánek (in the name of all co-authors)

Reviewer 2 Report

Comments and Suggestions for Authors

Peer review / Report for a manuscript Plants-2552123

This manuscript (authors Martin Kulhánek, Dinkayehu Alamnie Asrade, Pavel Suran, Ondřej Sedlář, Jindřich Černý, Jiří Balík) submitted to the journal Plants MDPI attempts to discuss new ways in plant nutrition based on the lessons of history.

On the one hand, there is some constructive potential in the concept of this manuscript.

On the other hand, there are many questions about its compliance with the scientific requirements and, accordingly, the possibility of its publication in the future.

It is quite obvious that the manuscript is absolutely uncompleted conceptually and, accordingly, not ready for publication.

In its current form, it is not possible to separate any of its positive aspects (although they may exist), since everything is blocked by a completely unsatisfactory approach to setting goals and objectives, as well as their implementation.

Definitely, at this stage, aspects of conceptualization, analysis, discussion and formation of conclusions in the manuscript are done very, very superficially.

More specific remarks, comments, and suggestions on this manuscript:

1. The title looks interesting, but definitely needs to be optimized.

There are 1) repetition of words, and 2) it is unnecessarily cumbersome.

2. The abstract is in need of significant revision: both stylistically and conceptually.

Authors need to express their thoughts more clearly and logically.

It is necessary to focus primarily on the novelty of the results of the work and the specific contribution of the authors to the field under study.

Lots of extra words there.

There is no special need for general words and well-known truths.

It is necessary to exclude statements that have a declarative-like character.

It is absolutely necessary to bring all this into a systemic form.

3. One gets the impression of certain confusion in the implementation of the concepts discussed in the manuscript, as well as in the presentation of the text of the manuscript as a whole and its individual parts.

It is necessary to carefully rework all this, providing clearly expressed logical connections.

4. Section Introduction has not been finalized and is not very informative.

It is necessary to more clearly substantiate the relevance of the manuscript and its novelty, more clearly formulate the purpose and objectives of the work, and also mention the previously published works of other authors that are similar in content and, if there are unmentioned ones, then their own ones too.

5. Perhaps a deeper and more systemic analysis of Table 1 is required there.

6. All main sections of the manuscript contain very little conceptual information, which is clearly insufficiently systematized. These must be deeply revised and carefully edited. It is necessary to present the information described there more clearly, more concisely, more specifically.

7. It is necessary to more carefully ensure the logical connection between the sections of the manuscript.

8. Regarding table 5: it is felt that there is not much additional information compared to the previous article to consider the updated approach acceptable.

9. Deep revision of the entire manuscript should lead the authors to a more logical, complete, systemic and conceptual formulation of the conclusions of the theoretical study and conclusion.

10. What is fundamentally new discovered by the authors as a result of the conducted, albeit theoretical, research?

11. Authors must be absolutely sure that this manuscript and its separate parts are published for the first time.

Thus, a significant revision of the manuscript by the authors, a deep revision of the entire text and significant re-editing of it are absolutely necessary.

Comments on the Quality of English Language

Extensive editing of English language required

Author Response

Dear Reviewer 2,

We are very grateful for your valuable review. Firstly, we have to apologize that we did not see some mistakes you pointed out before the manuscript was submitted. Because of that, we are very glad that you give us the chance to resubmit the corrected version of our manuscript.

Here is the point by point response to your remarks. For better orientation, your comments are included as well and highlighted in italics. Our response is always under your comment.

1. The title looks interesting, but definitely needs to be optimized. There are 1) repetition of words, and 2) it is unnecessarily cumbersome.

1. The title was changed due to the removing of repeating phrase on: Start of 21st Century in plant nutrition - new ways in based on the lessons of history – a review

2. The abstract is in need of significant revision: both stylistically and conceptually. Authors need to express their thoughts more clearly and logically. It is necessary to focus primarily on the novelty of the results of the work and the specific contribution of the authors to the field under study. Lots of extra words there. There is no special need for general words and well-known truths. It is necessary to exclude statements that have a declarative-like character. It is absolutely necessary to bring all this into a systemic form.

2. Abstract was almost completely rewritten. We specified the main objective, removed the extra words and general words. It is now written more structurally to point out the four most important reviewed groups of topics - i) long-term field experiments, ii) modern analytical methods, iii) new strategies in plant nutrition and iv) biotic and abiotic stresses.

The specific contribution of authors is mentioned under the study. They prepared the individual chapters related to their scientific expertise and I was responsible for completing the draft, which we discussed before submission.

3. One gets the impression of certain confusion in the implementation of the concepts discussed in the manuscript, as well as in the presentation of the text of the manuscript as a whole and its individual parts. It is necessary to carefully rework all this, providing clearly expressed logical connections.

3. The structure of manuscript was almost completely changed with the main focus on individual chapters connection. The idea is following:

Firstly is mentioned the role of long-term field experiments, because only long-term observation can give us the information valuable for agricultural practice. However, as the long term field experiments were established many years ago, it was possible to evaluate only some basic parameters (yields, bioavailable nutrients...).

Nowadays, it is possible to use the new evaluation techniques and technologies. Therefore, second part is dedicated to this topic. To prepare this part was very difficult, because to include all the new techniques is topic for one book. We therefore after discussion mentioned the techniques, which we are considering as the most important.

To do the analysis only for analysis is for nothing. It is necessary to implement the results in the agricultural practice. Because of that, third part is including the plant nutrition strategies, which is possible to implement according to the results of different analysis. We tried to focus even on these negative aspects of the new strategies (overestimating of biostimulants influence, trace elements in sewage sludge, ...).

To make study more complete, fourth part is dedicated to briefly description of the risks of biotic and abiotic stresses, where we are considering the salinity as the main barrier in modern pathways of plant nutrition.

We hope, that after the structural changes is the manuscript less confusing now.

4. Section Introduction has not been finalized and is not very informative. It is necessary to more clearly substantiate the relevance of the manuscript and its novelty, more clearly formulate the purpose and objectives of the work, and also mention the previously published works of other authors that are similar in content and, if there are unmentioned ones, then their own ones too.

4. The introduction was rewritten as well. We tried to write only very short introduction including the content of the review and its main focus to do not repeat parts mentioned in abstract and conclusion chapters. Furthermore, we added short introducing chapters to the main topics (new analytical techniques and modern fertilizing strategies) to improve the chapters connection and overall text flow. The works of other authors are therefore mentioned later in related subchapters.

5.Perhaps a deeper and more systemic analysis of Table 1 is required there.

5. We removed the title of chapter 3. "Overview of long-term field experiments with plant nutrition around the world", included in first version of our paper. The text from this chapter was modified and included to chapter 2. "Challenges of long-term observations related to plant nutrition" and the table 1. is now attached the chapter 2. to complete the information. We wanted to include the table only as without further description as an overview of long-term field experiments with links on related manuscripts. However, we can also include deeper analysis in text.

6. All main sections of the manuscript contain very little conceptual information, which is clearly insufficiently systematized. These must be deeply revised and carefully edited. It is necessary to present the information described there more clearly, more concisely, more specifically.

6. That was one of the manuscript deficiencies, which we should see before its submission. Thanks for this comment. As it is in response to your comment no. 3., we tried hard to improve the conceptualization and systematic.

7. It is necessary to more carefully ensure the logical connection between the sections of the manuscript.

7. We tried to improve the systematic as mentioned before (points 3. and 6.). We added the text parts accordingly to connect the chapters and subchapters to each other.

8. Regarding table 5: it is felt that there is not much additional information compared to the previous article to consider the updated approach acceptable.

8. Sorry for the wrong expression. The table was mostly modified. The update comprises only in including four new experiments (2 with Trichoderma harzianum, 1 for Pseudomonas sp. and 1 for Paenibacillus mucilaginosus). We did only this slight update, because the actual results does not show much new and we changed the information in the table description accordingly. However, we can include more new results into the table.

9. Deep revision of the entire manuscript should lead the authors to a more logical, complete, systemic and conceptual formulation of the conclusions of the theoretical study and conclusion.

9. We completely agree. With the revision of the main body of the manuscript we corrected the conclusions accordingly.

10. What is fundamentally new discovered by the authors as a result of the conducted, albeit theoretical, research?

10. Generally we have to say that we discovered nothing absolutely new. However, we see the "novelty" in following:

11. i) Actual results and even recommendation for the practice origin mostly from the pot experiments with regulated conditions. Therefore, it is easy to obtain significant results. Our study highlight that to get reliable results it is necessary to establish or evaluate the field experiments, which are laborious, expensive and with no immediately obtained results. 

11. ii) We included the critical look on the modern plant nutrition strategies (even in conclusions), e.g. overestimating of biostimulants effectiveness, risk elements in waste materials or negative aspects of plant breeding for higher nutrients acquisition. We think there is a lack of publications which are indicating these facts.

iii) Our study cannot be complete, because a lot of nowadays possibilities. However, most of the scientific studies is only monothematic (focused on only one nutrition strategy). We wanted to point out that only the combination of different systems can be effective.

Although these facts are absolutely not new, we are feeling that they are forgotten and should be repeatedly mentioned.

11. Authors must be absolutely sure that this manuscript and its separate parts are published for the first time.

11. Due to study of lot of materials, we were sure influenced by some of them and followed their ideas. However, me personally and I hope even all co-authors can state that all information obtained from other publications are properly cited.

Manuscript was checked with the native speaker to improve the English. Some changes were made even according the other two reviews.

Attached you can find the clean version as well as tracked version of revised manuscript.

We hope that the quality of the manuscript improved after the revisions and even if it will not be acceptable, your review showed us a way how to prepare the publications in the future.

Yours sincerely

Martin Kulhánek (in the name of all co-authors)

Reviewer 3 Report

Comments and Suggestions for Authors

Abstract

Line 13, “Should”, must be removed.

Line 17 in specific time revised to in a specific time.

Line 23 dGPS writes the complete form. Similarly, check all the paper and eliminate the same errors.

Introduction: The main conclusive paragraph at the end of the introduction still needs to be explained. What is the importance of this investigation and your target goals? 

Line 210, please revise to Schmidt et al. {90}, line 246, 261. Also, add space to the Mawodza and Thompson, and check the paper throughout for all mistakes……

Line 271: What does 22 mean?

Lines 356 and 365 remove one bracket. Check all the manuscripts for the same mistake.

Line 403 the size of the {201} is bigger than others

Line 416, please add space.

Line 533, please remove the space between the bracket and CO2

The table format should be the same and need more revision.

Line 892 add space to after (release)……………

Please check all the references cited in the text and vice-versa.

Comments on the Quality of English Language

Author Response

Dear Reviewer 3,

Thanks for your time you spend with the evaluation of your manuscript and for your valuable comments. We corrected all the typing errors you pointed out and checked the document thoroughly to avoid other ones.

Abstract and introduction were almost completely rewritten and we checked all of the references to be cited properly.

Corrected manuscript was checked by native speaker to improve the English language.

You can see all changes in tracked version of the manuscript. We are also attaching the clean version for better orientation.

Thanks again for your valuable review

Yours sincerely

Martin Kulhánek (in the name of all co-authors)

Round 2

Reviewer 2 Report

Comments and Suggestions for Authors

Dear Authors,

I read the second version of the manuscript with great interest.

I would like to say that I definitely welcome your approach based on a high-concept, deep and global analysis of current scientific challenges under setting goals and implementing the objectives of the manuscript.

I don't want to be too thorough, but I strongly recommend that you revise the manuscript again in accordance with previous comments.

It is highly advisable to do this on all counts.

I would like to see this manuscript implemented as much as possible in accordance with the principles of scientific validity and a systematic approach.

Comments on the Quality of English Language

Could be better.

Author Response

Dear Reviewer,

We are very grateful for the opportunity to revise the manuscript once more. We did the changes according to your previous review, which were not considered before.

1) The title was changed once more to be not so cumbersome on: "Plant nutrition - new methods based on the lessons of history - a review.

2) We added some relevant literature sources into the section introduction and changed whole introduction accordingly.

3) The analysis of the results published from long-term field experiments mentioned in table 1. was included under the table. The aim was to highlight that only basic parameters (yield, crop rotation, carbon storage, bioavailable nutrients content) are usually evaluated. However, nowadays equipment allow us to go more into the details.

4) Table 5. with biostimulants was updated. There are now included 7 more experiments as compared to previous version published by Holečková et al. (2017)

Whole manuscript was sent to PRS (https://www.proof-reading-service.com) for the academic proof-reading as well as for the general improvement of text flow and overall conciseness and completeness.

Attached you can find the tracked version of the manuscript. Almost one third of previous version was changed so we attached clean version as well for better transparency.

We hope we fulfilled all of your requests and we are of course open for further scientific discussion. If we have not included some changes, this is more likely due to a misunderstanding, for which we apologize and we are ready to incorporate any further comments.

Yours sincerely in the name of al co-authors

Martin Kulhánek

Reviewer 3 Report

Comments and Suggestions for Authors

The manuscript has been extensively improved according to the comments.

Author Response

Dear Reviewer,

Thanks a lot for positive evaluation of our manuscript. We did some more changes according to the 2nd reviewer's valuable recommendations. Furthermore, manuscript was academically proof-readed with  PRS (https://www.proof-reading-service.com) for the general improvement of english, text flow and overall conciseness and completeness.

We hope that overall quality of the manuscript was improved so far to be suitable to publish in Plants-Basel journal.

Attached you can find tracked and clean version of revised manuscript.

Yours sincerely in the name of al co-authors

Martin Kulhánek

Round 3

Reviewer 2 Report

Comments and Suggestions for Authors

What might be noted in the third round of review.

The authors generally did a lot of work - both before submitting the manuscript and as part of its editing after the comments and comments made.

If, in its current state, this is all that the authors are willing to do to improve the manuscript, then one could certainly recommend the manuscript for publication.

At the same time, even if someone considers this too picky on my part, it is clear that the questions and problems raised during the first round of reviewing still remain unrealized by the authors fully and, above all, in a conceptual context.

Definitely, coupled with the efforts already made by the authors to improve this manuscript, I would welcome further critical views of the authors on their work and appropriate rethinking in accordance with first peer review comments.

This manuscript definitely still needs conceptual development.

Comments on the Quality of English Language

It could be better.

Author Response

Dear Reviewer, Dear Editors,

We are very grateful for your time you spent with our manuscript. We tried to do the best to do the changes according to your request, it means mainly the change of the conceptualization.

To make the concept and structure of our manuscript more transparent, we re-arranged it in more chapters.

The concept of our manuscript is now following:

After short introduction (sect. 1) follows the chapter describing the long-term field experiments as a source of reliable background for modern analysis (sect. 2).

Section 3 was shortened on new instrumental analytical techniques only to highlight the equipment, which can be actually used for plant nutrition analysis. Furthermore, the order of section 3.1. and 3.2. was changed to be more chronologic. Section 3.1. is now describing the spectrometric and chromatographic methods, which are not new, but made big step forward in last two decades. Section 3.2. comprises the brand-new methods (tomography, magnetic resonance and X-ray fluorescence)

Section 4. follows with the newly investigated environments, where the previously mentioned analytical methods can be used - it means plant rhizosphere (4.1.), microbial communities and their enzymatic activities (4.2. and 4.3.) as well as for the evaluation of nutrients relationship (4.4.).

The section 5. is now dedicated to the data evaluation with the aim to point out that without adequate statistical analysis, the potential of "expensive" data reached with new techniques can be easily lost.

Section 6. follows with the modern ways of plant nutrition, where only the section and subsection numbering was changed (not the text part).

The same is valid for section 7. This should highlight actual risks represented mainly with different environmental stresses, which can strongly limit the effectiveness of some new fertilizing strategies.

Everything is summarized in conclusion chapter, where we tried to highlight even the negative aspects of new fertilizing methods, often omitted in the scientific publications.

Briefly, the concept of our manuscript is following now:

It should inform the reader about new analytical techniques and newly investigated environments, where the value of obtained data can be strongly supported with using them to evaluation of long-term field experiment.  Based on these (properly evaluated) results are subsequently developed and improved new fertilizing strategies mentioned in our manuscript. However, it is necessary to take care about global changes as well as consider negative aspects of newly developed fertilizing strategies.

Further changes:

Abstract was changed accordingly to the new structure of first sections.

The first paragraph of section 2. was shortened, because of repeating of obvious information.

Short introductions were added to the main sections to improve the continuity.

All above mentioned changes are visible in attached tracked version.

Furthermore, due to the repeated reading we found some typing errors and not clear expressions. We did the corrections accordingly directly into the clean version.

At the end we want to really thank you for your time you spent with our manuscript. We really did not consider your comments to be too picky and we tried to do our best to improve its quality. Each constructive review as yours is pushing us forward due to the new ideas and visions.

Yours sincerely

In the name of all co-authors.

Martin Kulhánek